# Smooth Dynamic Cutoffs for Machine Learning Interatomic Potentials

**Kevin Han**[1]  **Haolin Cong**[1]  **Bowen Deng**[2]  **Amir Farimani**[1]

## Abstract

Machine learning interatomic potentials (MLIPs) have proven to be wildly useful for molecular dynamics simulations, powering countless drug and materials discovery applications. However, MLIPs face two primary bottlenecks preventing them from reaching realistic simulation scales: inference time and memory consumption. In this work, we address both issues by challenging the long-held belief that the cutoff radius for the MLIP must be held to a fixed, constant value. For the first time, we introduce a **dynamic** cutoff formulation that still leads to stable, long timescale molecular dynamics simulation. In introducing the dynamic cutoff, we are able to induce sparsity onto the underlying atom graph by targeting a specific number of neighbors per atom, significantly reducing both memory consumption and inference time. We show the effectiveness of a dynamic cutoff by implementing it onto 4 state of the art MLIPs: MACE, Nequip, Orbv3, and TensorNet, leading to **2.26x** less memory consumption and **2.04x** faster inference time, depending on the model and atomic system. We also perform an extensive error analysis and find that the dynamic cutoff models exhibit minimal accuracy dropoff compared to their fixed cutoff counterparts on both materials and molecular datasets. All model implementations and training code will be fully open sourced.

## 1. Introduction

In recent years, machine learning interatomic potentials (MLIPs) trained on quantum chemical calculations such as density functional theory (DFT) (Argaman & Makov, 2000) or coupled clustering (CC) (Bartlett & Musiał, 2007) have been shown to be extremely useful in both materials discov-

[1]Carnegie Mellon University [2]Massachuessets Institute of Technology. Correspondence to: Kevin Han <kevinhan@cmu.edu>.

*Proceedings of the 43rd International Conference on Machine Learning*, Seoul, South Korea. PMLR 306, 2026. Copyright 2026 by the author(s).

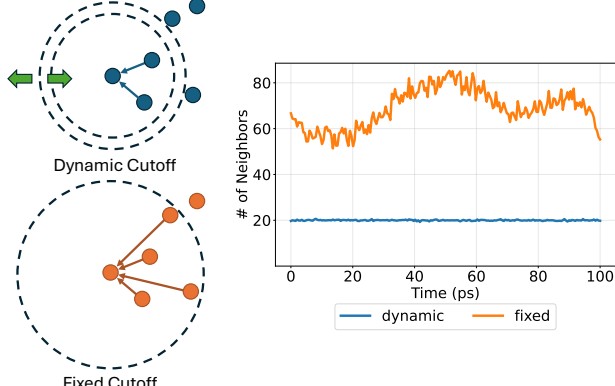

*Figure 1.* We introduce a dynamic cutoff function which induces graph sparsity onto the underlying atom graph while maintaining simulation stability. The dynamic cutoff function calculates a dynamic radius that targets a specific number of atoms to be within the dynamic radius while pruning the rest. This sparsification method leads to up to 2.26x reduction in memory consumption and up to 2.04x reduction in inference time depending on the model and atomic system.

ery and drug discovery applications (Deringer et al., 2019). At its most fundamental level, MLIPs, typically based on graph neural networks, input an atomic system (atom coordinates and types) and output the energy of the overall system as well as the forces applied upon each atom (Jacobs et al., 2025). The energy and forces are then used to drive simulations such as molecular dynamics, which integrate Newton's laws of motion to determine the positions of the atoms at the next timestep (Larsen et al., 2017). However, due to the need for stable integration of the fast oscillatory motion of atoms, the integration timestep is set at the femtosecond level, requiring millions to billions of MLIP inferences for a single simulation (Hollingsworth & Dror, 2018). As a result, even small increases in inference time can lead to days of additional simulation time. Furthermore, high-throughput drug and materials discovery workloads typically require simulating large atomic systems or large amounts of smaller atomic systems, both of which are bottlenecked by overall GPU memory (Harvey & De Fabritiis, 2012). These two requirements underscore the necessity for MLIPs that are both fast in inference and consume minimal GPU memory.

There have been a few works focused on increasing MLIP

inference and reducing memory consumption. Lee et al. (2025), Tan et al. (2025), Geiger et al. (2024), and Bharadwaj et al. (2025) implement custom CUDA kernels into specific models to accelerate expensive equivariance calculations. Kong et al. (2025) attempt to remove entire layers from the MLIP, but find that energy and force errors increase significantly. Musaelian et al. (2023) introduce Allegro, a fast and memory efficient MLIP. However, Allegro is restricted to only modeling local, short-range interactions, which may not be sufficient to accurately capture complex atomistic dynamics for many systems (Anstine & Isayev, 2023; Leimeroth et al., 2025).

In graph-based MLIPs, the underlying atom graphs are constructed by searching each atom's neighbors within a *preset, fixed* cutoff distance. One promising avenue for reducing inference time and memory consumption lies in restricting the maximum number of neighbors for each atom to the $n$ nearest neighbors – inducing graph sparsity to the atom graph (Kozinsky et al., 2023; Rhodes et al., 2025). Only including the $n$ nearest neighbors significantly reduces the number of total edges and arithmetic operations during model inference. The inclusion of specifically the $n$ *nearest* neighbors is grounded in the decaying interaction strength between two atoms as the distance between them increases, such that the nearest neighbors can capture the majority of the interactions (Jones, 1924). However, enforcing a maximum neighbor restriction explicitly leads to discontinuities in the potential energy surface (PES), as seen in Figure 2, resulting in grossly unstable molecular dynamics simulation (Fu et al., 2022; 2025c).

In this work, we present a smooth and physically stable variant of the maximum neighbor restriction by challenging the long-held notion that the cutoff radius for an MLIP must be fixed. We propose a dynamic cutoff function that maintains the higher-order differentiability of the MLIP while making sure that the atoms within the dynamic cutoff roughly hold a target $n$ number of atoms.

We implement the dynamic cutoff on 4 popular and state-of-the-art MLIP models: MACE, Nequip, Orbv3, and TensorNet. Ablations on the MD22 (molecular systems) and MatPES (material systems) datasets show that **an MLIP trained with a dynamic cutoff results in minimal accuracy reduction while simultaneously consuming up to 2.26x less memory and up to 2.04x less inference time** depending on model and atomic system.

## 2. Related Work

Modern machine learning interatomic potentials (MLIPs) are typically graph-based neural networks trained on the energy and forces calculated from expensive first-principles, quantum mechanical methods (Musaelian et al., 2023;

Batzner et al., 2022; Batatia et al., 2022; Deng et al., 2023; Simeon & De Fabritiis, 2023; Fu et al., 2025a; Wood et al., 2025; Rhodes et al., 2025; Park et al., 2024). MLIP models are then used as surrogates for expensive quantum mechanical calculations for molecular dynamics simulations, enabling the study of atomistic systems ranging from battery cathode materials, to protein systems, to metal organic frameworks (Hollingsworth & Dror, 2018; Zhong et al., 2025; Greathouse & Allendorf, 2006). However, MLIPs are still computationally far from simulating atomistic systems of real-world scale within a reasonable amount of time (Han et al., 2025).

There is growing research interest in increasing MLIP inference speed and reducing memory consumption. For equivariant MLIPs such as Nequip, MACE, and SevenNet, custom CUDA kernels have been created for expensive tensor products and spherical convolutions – leading to nontrivial memory consumption reduction and inference speed gains (Lee et al., 2025; Tan et al., 2025; Geiger et al., 2024; Bharadwaj et al., 2025). Additionally, in efforts to reduce the overall FLOP count for inference, Kong et al. (2025) prune entire layers of some MLIPs; however, they notice a substantial few-fold increase in error after doing so. On the other hand, DistMLIP presents a distributed inference platform allowing various MLIPs to perform memory-efficient inference on multiple GPUs, leading to substantial inference speedup and simulation capacity increase (Han et al., 2025). Rhodes et al. (2025) notices that inducing sparsity over the underlying atom graph by instituting a maximum neighbor limit of the 20 nearest neighbors leads to a strong speed and memory improvement. However, along with Fu et al. (2025c), they show that the resulting MLIP has a non-smooth PES, as seen in Figure 2, leading to highly unstable and un-usable molecular dynamics simulation.

Fu et al. (2025b) and Hairer et al. (2003) note and prove that the stability, and subsequent usability, of the molecular dynamics simulation requires the PES to be at least twice-differentiable – with the numerical error bound on the energy drift of the simulation decreasing substantially with the order of differentiability of the PES.

There has been little prior work investigating the cutoff radius for MLIPs. Classical molecular dynamics, which don't involve machine learned models, have predefined fixed cutoffs depending on the elements of the system being simulated (Zuo et al., 2020). Takamoto et al. (2022) introduce an MLIP with a different fixed cutoff for each layer of the model (e.g. 3Å for the first layer, 4Å for the second, etc.).

In this work, **our goal is to reap the benefits of inducing graph sparsity while still maintaining stable simulation** To this end, we challenge the long-held belief that MLIPs must maintain a fixed cutoff radius in order to maintain a smooth PES, and subsequently, support a stable molecular

dynamics simulation. We design a dynamic cutoff function that maintains the higher-order differentiability of the MLIP while maintaining the number of atoms within the cutoff to hover around a pre-specified target number of neighbors. This allows us to induce graph sparsity and set a "soft maximum neighbor" count while still maintaining smoothness of the energy surface as well as stable molecular dynamics simulation. A depiction of the dynamic cutoff can be found in Figure 1.

## 3. Dynamic Cutoff Formulation

Let $v$ be a node, representing an atom, and $N_v$ be the set of incoming neighboring nodes to $v$ within a hard cutoff radius of $h$. Let $u$ be another node such that $u \in N_v$, and let $r_{uv}$ describe the distance of the edge from atom $u$ to atom $v$.

For all $u \in N_v$, we define a rank $R_u$ where

$$R_u = \sum_{t \in N_v \setminus \{u\}} \left[ S(\alpha * (r_{uv} - r_{tv})) p\left(\frac{r_{tv}}{h}\right) \right] \quad (1)$$

where $S$ is the sigmoid function, $\alpha \in \mathbb{R}^+$, and $p : \mathbb{R} \to \mathbb{R}$ is a polynomial envelope function that smoothly decays to 0 when $\frac{r_{tv}}{h}$ goes to 1. One mathematical form of $p$ was introduced in Gasteiger et al. (2020), defined as

$$p(x) = 1 - \frac{(n+1)(n+2)}{2}x^n$$
$$+ n(n+2)x^{n+1} - \frac{n(n+1)}{2}x^{n+2}$$

for some $n \in \mathbb{N}^+$ where $n \geq 3$. Note that, by design, $p(1) = 0$, $p'(1) = 0$, and $p''(1) = 0$. Equation 1 can be interpreted as the soft rank of node $u$ with respect to all other neighbors $t \in N_v$. The sigmoid behaves as a smooth "indicator" function for the relative ordering of $u$ and $t$ while the summation over indicators leads to a smooth ranking of $u$ within $N_v$. The polynomial envelope function $p$ is used to prevent a sudden jump in all rankings when an atom in $N_v$ leaves the hard cutoff $h$. $\alpha$ determines the sharpness of the indicator function while $n$ determines the steepness of the polynomial envelope. These values are typically fixed at 10 and 50, respectively. We perform the rankings in Equation 1 for all nodes $v$ in the atom graph $G$. Additional discussion on neighbor ranks can be found in Appendix E.

Next, we use the PDF of the normal distribution to define a smooth weighting function over ranks. We define $\omega : \mathbb{R} \to \mathbb{R}$ as

$$\omega(x) = \frac{1}{\sigma\sqrt{2\pi}} e^{-\frac{1}{2}\left(\frac{x-\mu}{\sigma}\right)^2} \quad (2)$$

where $\mu, \sigma \in \mathbb{R}^+$. $\omega$ is interpreted as a symmetric weighting function for each $u \in N_v$ based on $u$'s ranking. As a result, $\mu$ determines the target average number of neighbors we

would like to have within the cutoff while $\sigma$ determines the tolerance, or variation, around the average number of neighbors. We find $\sigma$ of 4 to work well in maintaining minimal variation across diverse chemistries.

Finally, we calculate the dynamic cutoff, $c_v$ for node $v$ using a weighted average as follows:

$$c_v = f(N_v) = \frac{\left[\sum_{u \in N_v} \omega(R_u) p\left(\frac{r_{uv}}{h}\right) r_{uv}\right] + h\epsilon}{\left[\sum_{u \in N_v} \omega(R_u) p\left(\frac{r_{uv}}{h}\right)\right] + \epsilon} \quad (3)$$

where $\epsilon$ refers to a small value such as 1e-4 and is used to maintain numerical stability. When the final neighboring atom $u$ leaves the hard cutoff $h$, the dynamic cutoff pushes towards $h$ via the $h\epsilon$ in the numerator of the weighted sum. By design, $c_v$ can range from 0 to $h$ depending on the locations of its neighbors $N_v$. The $p(\frac{r_{uv}}{h})$ terms in the numerator and denominator of the weighted sum maintain smoothness in the weighting function when an atom $u$ leaves the hard cutoff, $h$. Furthermore, provided two atoms $a$ and $b$, depending on the local density of the atomic system, it is possible for $a$ to have an edge to $b$ but $b$ to not have an edge to $a$. In other words, implementing a dynamic cutoff, or any maximum neighbor implementation, leads to a potentially asymmetric adjacency matrix depending on the atomic configuration.

Empirically, we find the dynamic cutoff function to robustly and reliably maintain, on average, the target number of neighbors $\mu$ within the dynamic cutoff, with little variation. However, due to the smooth nature of the dynamic cutoff, there is no guarantee that each atom will have exactly $\mu$ neighbors at all points during simulation.

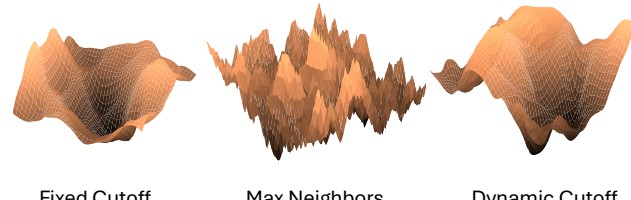

Fixed Cutoff     Max Neighbors     Dynamic Cutoff

*Figure 2.* The potential energy surface (PES) of a fixed cutoff, max neighbors, and dynamic cutoff TensorNet model on a random system. The fixed cutoff baseline and dynamic cutoff energy surfaces exhibit smooth characteristics while the energy surface resulting from setting a strict maximum neighbor is highly jagged and unsmooth. The PES associated with the maximum neighbor model leads to highly unstable and un-usable molecular dynamics simulation while the the fixed cutoff and dynamic cutoff models lead to stable simulation.

In practice, gradients must flow from the energy prediction to the atomic positions through the cutoff as well as the

model weights. An initial neighbor list construction step is performed using a standard fixed cutoff at $h$ before the dynamic cutoff $c_v$ is used to prune out neighbors beyond $c_v$ in the graph and serve as the new cutoff for weighting incoming messages.

In Figure 2, we show the PES of a traned TensorNet fixed cutoff (baseline) model, dynamic cutoff model, and naive max neighbors version. The dynamic cutoff and fixed cutoff models show smooth energy surfaces, while the strict max neighbors model leads to a highly jagged, unusable, and unphysical surface.

As shown by Hairer et al. (2003) and Fu et al. (2025a), the bounds on the energy drift for a simulation is a function of the order of differentiability of the PES, with second-order differentiability being the minimum requirement (i.e. forces must be differentiable) for stable simulation. We provide a proof of the second-order differentiability of the dynamic cutoff as well as corroborate its simulation stability by performing long timescale simulations across both molecular and high temperature material systems in Section 4.4.

**Theorem 3.1.** *Given an arbitrary atomic system with arbitrary movement of atoms as well as an MLIP $f$ that is at least second-order differentiable, the graph constructed using the cutoffs from the dynamic cutoff function $C$ always results in an energy surface that is second-order differentiable with respect to the position of each atom.*

*Proof.* See Appendix A for the full proof. □

## 4. Results

### 4.1. Experimental Setup

We implement the dynamic cutoff function into 4 popular, state-of-the-art MLIP models widely used for stable molecular dynamics simulation: MACE, Nequip, Orbv3, and TensorNet. MACE, Nequip and TensorNet are equivariant models while Orbv3 is invariant.

To compare the training performance between the dynamic cutoff and fixed cutoff models, we train both versions of each of the 4 models on the MatPES and MD22 datasets. Notably, all models are trained using the same hyperparameters between the fixed cutoff and dynamic cutoff versions. We did not perform any hyperparameter tuning and primarily rely on the hyperparameter defaults of each model for each dataset. We report comparisons between the fixed cutoff and the dynamic cutoff versions of each model without changing any other setting. All ablations are performed on smaller versions of each of the 4 models, ranging from 200k to 500k parameters. All hyperparameter details are outlined in Appendix B. For material systems, we choose the target average number of neighbors, $\mu$, to be 40, and for molecular systems, we set $\mu$ to be 20. Empirically, we

find these settings to sufficiently capture the neighboring atomic environments while still providing strong inference speed acceleration and memory reduction when simulating real-world systems.

### 4.2. Molecules

We trained each of the 4 models on all 7 large molecules in the MD22 dataset (Chmiela et al., 2023). The MD22 dataset consists of 7 molecular dynamics trajectories calculated using high-fidelity DFT. We use the same training and validation splits outlined in the original MD22 paper.

The resulting energy and force mean absolute errors (MAEs) on the validation set can be found in Table 1. Energy error and force error units are in meV/atom and meV/Å respectively. Although the dynamic cutoff version of the model oftentimes shows larger errors, the errors are typically bounded within 0.1-0.2 meV/atom for energies and a 2-3 meV/Å for forces depending on the system and model.

### 4.3. Materials

We also trained each of the 4 models on the MatPES-r2scan dataset, a challenging foundational materials dataset consisting of ∼400,000 structures from room temperature molecular dynamics simulation (Kaplan et al., 2025). We train on 80% of the data and validate on the other 20%. Validation MAEs can be found in Table 2. Energy errors are in meV/atom while force errors are in meV/Å. Note that we perform no hyperparameter tuning between the fixed and dynamic cutoff versions of the model. The dynamic cutoff versions of the model show minimal error increase compared to the fixed versions of the model. The target number of neighbors, $\mu$, is set to 40 for this dataset.

### 4.4. Simulation stability

The naive solution to implementing a maximum neighbor limit, setting the cutoff for node $v$ to the distance of the $n+1$st neighbor, introduces non-differentiable cusps within the PES. These cusps lead to a highly unstable and unusable simulation, which is described in Fu et al. (2025a) and Rhodes et al. (2025). Our dynamic cutoff, on the other hand, is provably second-order differentiable while still targeting the specified target number of neighbors, $\mu$.

In Figure 3, we demonstrate the simulation stability across all 4 models in which we implement the dynamic cutoff function. We plot the energy drift of the total energy in meV/atom over a 100 ps molecular dynamics (MD) simulation under constant number of particles, volume and energy (NVE ensemble) – the classic way to test energy conservation of a given force field (Fu et al., 2022). The NVE-MD simulations were performed on a double walled nanotube system from the MD22 dataset (left of Fig. 3) and

*Table 1.* Energy and force MAEs for both the fixed cutoff and dynamic cutoff versions of the MACE, Nequip, Orbv3, and TensorNet models trained on the MD22 dataset. Note that these are smaller versions of the models (between 200k and 500k parameters, see Appendix B for details). Units are in meV/atom and meV/Å for energy and forces respectively. We perform no hyperparameter tuning between the fixed and dynamic cutoff versions of the models. The dynamic cutoff model energies are consistently within 0.1-0.2 meV/atom of the fixed cutoff models. The dynamic cutoff forces are also consistently within 2-3 meV/Å of the fixed cutoff models. The target neighbor of neighbors, $\mu$ was set to 20 for the MD22-trained models.

| MODEL | CUTOFF | | AC-ALA3-NHME | AT-AT | AT-AT-CG-CG | DHA | BUCKYBALL | STACHYOSE | DWNT |
|---|---|---|---|---|---|---|---|---|---|
| MACE | FIXED | ENERGY | 0.217 | 0.212 | 0.243 | 0.247 | 0.434 | 0.239 | 0.247 |
| | | FORCE | 14.28 | 17.35 | 24.37 | 13.95 | 24.88 | 20.68 | 35.60 |
| | DYNAMIC | ENERGY | 0.264 | 0.286 | 0.308 | 0.277 | 0.265 | 0.269 | 0.291 |
| | | FORCE | 15.03 | 20.13 | 27.26 | 15.05 | 27.71 | 21.81 | 41.22 |
| NEQUIP | FIXED | ENERGY | 0.396 | 0.438 | 0.785 | 0.370 | 0.392 | 0.681 | 0.434 |
| | | FORCE | 17.30 | 21.77 | 29.92 | 13.79 | 27.75 | 24.89 | 40.59 |
| | DYNAMIC | ENERGY | 0.607 | 0.867 | 0.971 | 0.607 | 0.373 | 0.823 | 0.399 |
| | | FORCE | 17.52 | 22.11 | 30.35 | 14.53 | 30.35 | 26.45 | 47.02 |
| ORBV3 | FIXED | ENERGY | 0.347 | 0.265 | 0.270 | 0.229 | 0.564 | 0.457 | 0.282 |
| | | FORCE | 12.57 | 21.42 | 30.94 | 14.59 | 22.94 | 16.29 | 23.81 |
| | DYNAMIC | ENERGY | 0.477 | 0.347 | 0.278 | 0.245 | 0.390 | 0.520 | 0.283 |
| | | FORCE | 13.87 | 22.07 | 30.78 | 15.44 | 30.35 | 16.39 | 26.02 |
| TENSORNET | FIXED | ENERGY | 6.374 | 1.084 | 1.068 | 0.604 | 3.29 | 1.259 | 6.18 |
| | | FORCE | 25.93 | 43.40 | 59.84 | 25.74 | 18.33 | 40.96 | 33.40 |
| | DYNAMIC | ENERGY | 6.071 | 1.037 | 1.143 | 0.742 | 3.30 | 1.201 | 6.29 |
| | | FORCE | 27.44 | 43.50 | 61.79 | 26.11 | 20.28 | 43.46 | 40.76 |

*Table 2.* The energy (meV/atom) and force (meV/Å) MAEs for the MACE, Nequip, Orbv3, and TensorNet models on the challenging MatPES-r2scan dataset. Note that these are smaller versions of the models (between 200k and 500k parameters, see Appendix B for details). The target number of neighbors, $\mu$, is 40 for all dynamic cutoff models.

| MODEL | METRIC | FIXED | DYNAMIC |
|---|---|---|---|
| MACE | ENERGY | 30 | 31 |
| | FORCES | 173 | 190 |
| NEQUIP | ENERGY | 59 | 70 |
| | FORCES | 167 | 168 |
| ORBV3 | ENERGY | 67 | 68 |
| | FORCES | 176 | 177 |
| TENSORNET | ENERGY | 39 | 43 |
| | FORCES | 152 | 160 |

a LiFePO4 supercell consisting of 224 atoms (right of Fig. 3). For the simulation of the double walled nanotube, we follow Fu et al. (2025a) and first perform 500 steps of FIRE relaxation before randomly initializing the temperature of the system to 400K and running 100 ps of NVE simulation (Bitzek et al., 2006). For the simulation of LiFePO4, in order to induce a large number of neighbor switches to stress test the dynamic cutoff stability, we initialize velocities to 3000K, perform NVT at 3000K for 10 ps, and then run NVE for 100 ps. All models show no large-scale, systemic drift across the entire simulation – corroborating the stability of the dynamic cutoff.

## 4.5. Memory and inference time reduction

The goal of the dynamic cutoff is to reduce memory consumption and inference time of the simulation. This leads to increased throughput of high throughput materials/drug screening workflows, accelerated simulation times, and increased simulation cell size. We benchmark the models trained in Section 4.2 and Section 4.3 on 2 real world systems – a metal organic framework (MOF) material supercell consisting of 7500 atoms as well as a large biomolecule consisting of 16926 atoms. The MOF used was H4Pb(C2O3)2 while the protein used was 1ADO from the protein databank, a rabbit muscle protein (Bank, 1971). The target number of neighbors, $\mu$, was 40 and 20 respectively, aligning with the choices of $\mu$ for the MatPES (material) and MD22 (molecule) datasets. The timing and memory consumption results are found in Figure 4. Due to TensorNet's higher memory requirements, the MOF and protein systems were cut into 1/3rd in order to perform benchmarking. For fair comparison, the overall edge density was maintained with the original systems. To explicitly measure inference time acceleration of the model, we follow Wood et al. (2025) and Rhodes et al. (2025) and exclude graph construction time in the inference time benchmark, which is negligible for most models.

Using the dynamic cutoff, the total number of edges in the material and molecular systems are reduced from 677000 to 297000 and 805040 to 588141 respectively. Although Equation 1 scales quadratically with the number of neigh-

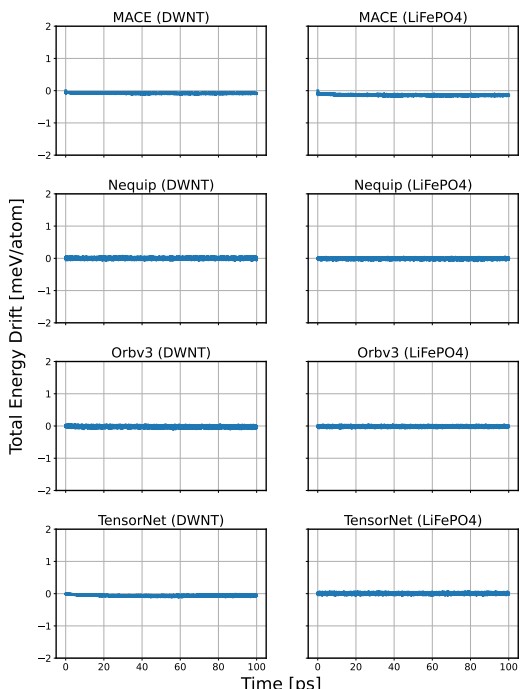

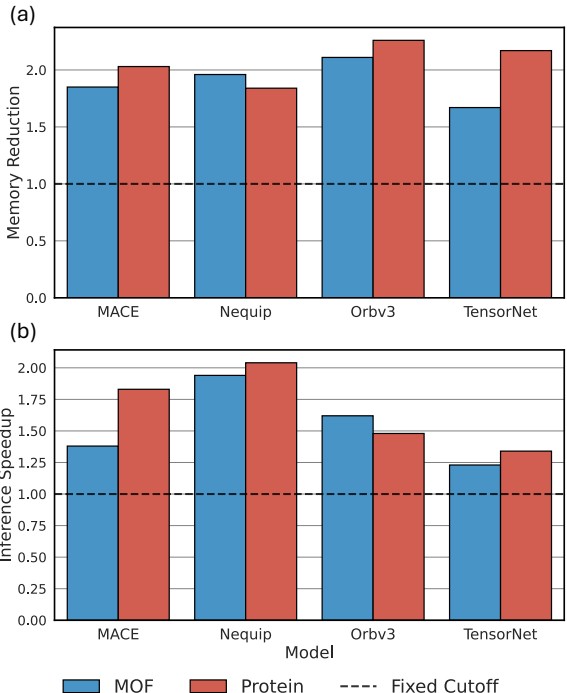

*Figure 3.* Plots showing the total energy drift in meV/atom of a 100 ps NVE molecular dynamics simulation using the dynamic cutoff models. DWNT refers to the double walled nanotube system from the MD22 dataset. LiFePO4 refers to an LiFePO4 supercell simulated at 3000K in order to stress test the dynamic cutoff by inducing a large number of neighbor switches throughout the simulation. Besides standard perturbations within the energy due to numerical integration, there exists no systemic drift throughout the entirety of the simulation.

*Figure 4.* (a) A plot of the memory reduction (higher is beter) of the MACE, Nequip, Orbv3, and TensorNet models when using a dynamic cutoff compared to a fixed cutoff for a periodic material (MOF) and biomolecular protein system. The target number of neighbors, $\mu$ is 40 and 20 for the material and molecular systems respectively. (b) A plot of the inference time speedup (higher is better) for each of the 4 models on the two systems relative to the fixed cutoff inference time.

bors, the number of neighbors is effectively bounded by the maximum density of naturally occurring atomic systems of interest (Wood et al., 2025; Rhodes et al., 2025). As a result, the time and memory required to calculate the dynamic cuotff are consistently less than 1% of the total inference time and memory consumption. Most computation time is spent during forward convolutions; however, nonzero overhead within initial graph featurization and aggregation exists, leading to suboptimal memory and inference time scaling with respect to the number of edges.

### 4.6. Error Analysis

Since the dynamic cutoff prioritizes the inclusion of the nearest neighbors, we perform an analysis on how the errors of the dynamic cutoff model increase when compared to its fixed cutoff variant. We take the TensorNet models trained on the MatPES dataset (in Section 4.3) and compare their per-element force errors on the validation set. We plot the the relative increases in per-element force MAEs with respect to the Clementi calculated atomic radii of the elements

in Figure 5 (Clementi & Raimondi, 1963). We observe that the largest electropositive atoms with the longest bond lengths in the MatPES dataset (potassium, rubidium, cesium, and actinium) have the highest increase in force errors. Furthermore, the general trend is for atoms with larger atomic radii, such as sodium and barium, to have larger increases in force errors compared to smaller elements such as helium or carbon. This is most likely a result of the dynamic cutoff being unable to fully capture all atoms within its hard cutoff, $h$, when there are less than $\mu$ atoms within the hard cutoff neighborhood. This behavior can be seen in Figure 8, where as the number of neighbors approaches and reduces below $\mu$, the number of atoms within the dynamic cutoff also begins to drop slightly. However, this issue can be alleviated by simply increasing the hard cutoff $h$ while maintaining $\mu$. The full plot outlining each of the per-element force errors can be found in Appendix F.

### 4.7. Choosing $\mu$

We show that increasing the target number of neighbors increases model accuracy in Figure 6. In the figure, we train several iterations of the same TensorNet model (456k parameters) using the same hyperparameters as Section 4.2

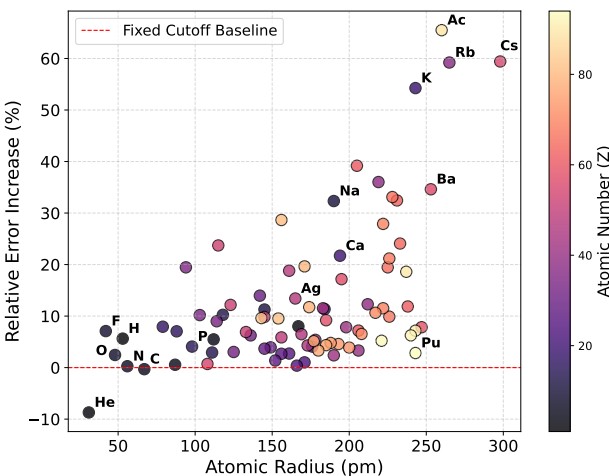

*Figure 5.* The relative force MAE increase of the dynamic cutoff TensorNet model trained on the MatPES dataset compared its fixed cutoff counterpart. The relative error increase is plotted with respect to the Clementi calculated atomic radius.

and 4.3. The hard cutoff $h$ was set to 6Å while the the target number of neighbors, $\mu$, was scaled from 5 to 40. We also include the accuracy of the model using a fixed cutoff of 6Å for comparison to a baseline. Both of the $\mu = 40$ models converge to their fixed radius baselines with the buckyball-catcher $\mu = 40$ model outperforming the fixed radius baseline. Understandably, increasing the average number of neighbors, $\mu$, directly correlated with more accurate models. The average number of neighbors under the 6Å fixed cutoff models was around 47 and 64 for the buckyball catcher and double walled nanotube systems respectively.

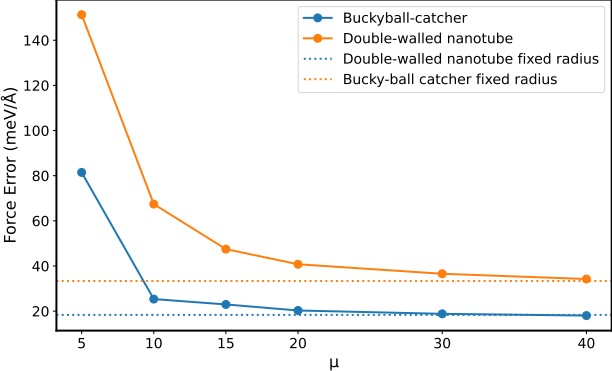

*Figure 6.* Relationship between force MAE (meV/Å) and target number of neighbors, $\mu$ on the double walled nanotube and buckyball catcher datasets from MD22. The models trained were TensorNet models with 456k parameters using the same hyperparameters as Section 4.2. Each TensorNet with different $\mu$ is trained separately.

## 5. Discussion

### 5.1. Dynamic Cutoff vs. Reducing Fixed Cutoff

A dynamic cutoff poses several advantages over simply reducing the fixed cutoff radius, especially for foundational MLIPs trained on a range of atomic systems that are diverse in atomic density (atoms/Å$^3$) and chemistries. Compared to a fixed cutoff set to the target number of neighbors for a specific system, a dynamic cutoff handles atomic density changes during a simulation more robustly. For example, a crack propagation simulation will have a void open up at a centralized location in the material (Mullins, 1982). The direct neighborhood around the void will see a significant atomic density decrease, leading a reduced fixed cutoff model to drop neighbors and lose accuracy for the critical atoms near the tip. A dynamic cutoff model, on the other hand, will be able to extend its cutoff to still maintain enough neighbors to accurately calculate energy and forces. Other examples of density changes during a simulation can be found in the formation of gas bubbles on surfaces, evaporation of liquid systems into their gaseous states, and physical expansion of lithium battery anodes during charging (She et al., 2016; Zhakhovskii & Anisimov, 1997; Toki et al., 2024). In essence, if the density of the atomic system changes even slightly throughout the simulation, a fixed cutoff model will drop neighbors crucial for accurate energy and force calculations while a dynamic cutoff model adjusts its cutoff radius to stably include the target number of neighboring atoms.

We illustrate this difference in a high temperature melting simulation of 864 atoms of copper at 4500K over 10 ps. We perform the simulation with a well-calibrated classical potential accounting for all pair-wise interactions under the NVT ensemble using a Langevin thermostat. We then plot the average number of neighbors at each timestep under a reduced fixed cutoff of 4.25Å (blue curve in Figure 7). This reduced fixed cutoff radius is chosen such that it averages to 20 neighbors per atom in bulk copper. We also plot the average number of neighbors for a dynamic cutoff with $\mu = 20$ (green curve in Figure 7). The average number of neighbors for the reduced fixed cutoff decreases significantly throughout the simulation while the dynamic cutoff aligns more closely with the unreduced fixed cutoff of 7Å.

Another advantage a dynamic cutoff model has over a reduced, fixed cutoff model involves the training of foundational MLIPs on a diverse array of atomic systems. A foundational MLIP with a fixed cutoff can result in an overwhelming amount of neighbors in atomically dense systems within the training data, such as alloys, and too few neighbors in atomically sparse systems. Dynamic cutoffs, on the other hand, reconcile this issue between dense and sparse chemical systems. We show this difference by training a TensorNet model with its fixed cutoff tuned to average 40

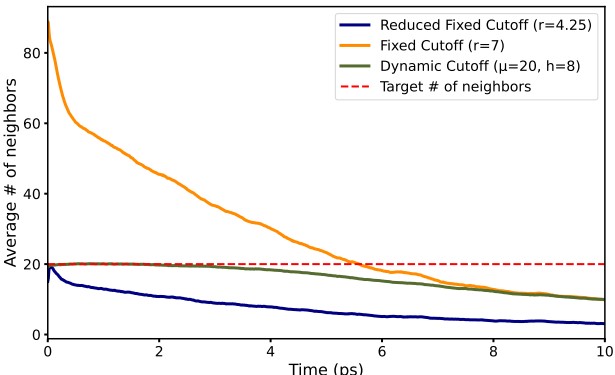

*Figure 7.* The average number of neighbors per atom for an 864 atom copper melting simulation at 4500K using a classical potential. We show that simply reducing a fixed cutoff radius to average 20 neighbors/atom in the bulk state (blue curve) is insufficient for capturing local neighborhoods and maintaining the desired number of neighbors in molecular dynamics simulation where the atomic density (atoms/$\text{Å}^3$) may fluctuate. Dynamic cutoffs (green) on the other hand, can expand their cutoff radii to include more neighbors as the atomic density decreases during the simulation. We include a fixed cutoff at 7Å (orange) to show the average number of neighbors in an unreduced fixed cutoff setting for this simulation.

*Table 3.* The energy and force MAEs (meV/atom and meV/Å) of a dynamic cutoff TensorNet ($\mu = 40$) trained on MatPES and a TensorNet with a reduced fixed cutoff calibrated to average 40 neighbors/atom in the training set. The training and architecture hyperparameters for both versions of the model match the TensorNet models trained in Section 4.3.

| CUTOFF TYPE | ENERGY | FORCES |
|---|---|---|
| DYNAMIC CUTOFF | 43 | 160 |
| REDUCED FIXED CUTOFF (5.4141Å) | 53 | 171 |

neighbors on the MatPES dataset (approximately 5.4141Å). We initialize the model using the same architecture and training hyperparameters as the TensorNet models trained in Section 4.3 and report the results in Table 3. The dynamic cutoff TensorNet results were taken directly from the MatPES experiments in Section 4.3. As seen from the table, the dynamic cutoff model can converge to lower energy and force error compared to the reduced cutoff model even though the overall average number of neighbors across the dataset is the same – highlighting how foundational potentials benefit strongly from using a dynamic cutoff over tuning a fixed cutoff that may not be optimal for a wide variety of atomic systems.

### 5.2. Neighbor Count

The current dynamic cutoff formulation represents a smooth way to target an average number of neighbors per atom. However, in order to adhere to simulation stability and second-order differentiability constraints, dynamic cutoffs are not able to maintain the exact target $\mu$ for all possible atomic configurations. Specifically, the primary extreme failure mode for a dynamic cutoff lies when all neighboring atoms are equidistant to a center atom. In this case, setting a spherical cutoff radius to any value will be forced to either include all of the neighbors or none of them. Otherwise, under reasonable choices of $\alpha$ and $\sigma$ in Equations 1 and 2, we find the dynamic cutoff to be highly capable of choosing the cutoff such that there are close to $\mu$ neighbors on average. In particular, the choice of $\alpha$ affects the strength in which $R$ distinguishes the ranking of each neighbor $u \in N_v$ from one another while the choice of $\sigma$ determines how "sharp" the average number of neighbors will revolve $\mu$.

### 5.3. Orthogonality

The dynamic cutoff induces sparsity onto the underlying atom graph. Therefore, the memory consumption decrease and inference acceleration lies orthogonal to speedups resulting from faster equivariance kernels and lead to multiplicative gains when both techniques are used in tandem (Bharadwaj et al., 2025; Geiger et al., 2024; Tan et al., 2025; Lee et al., 2025). Furthermore, model pruning techniques such as those presented by Kong et al. (2025) as well as the distributed inference platform, DistMLIP, presented by Han et al. (2025) are also orthogonal and would lead to multiplicative speedup and memory consumption decreases. For example, a custom CUDA kernel leading to 4x memory reduction and 4x inference time reduction combined with an 8 GPU setup from DistMLIP as well as a dynamic cutoff, would lead to a total of around 64x larger simulatable systems that should, assuming no overhead, run around 64x faster per atom-timestep. The implementation of a dynamic cutoff does not replace any of the previous acceleration techniques, but rather complements them and multiplies them instead.

## 6. Conclusion

In this work, we challenge the commonly held assumption that the cutoff radius of a machine learning interatomic potential (MLIP) must remain a fixed, constant value in order for molecular dynamics simulation to be stable. We introduce a dynamic cutoff formulation, a setting in which the cutoff radius is variable. We prove that our formulation is second-order differentiable and therefore leads to stable long timescale simulation. Using a dynamic cutoff to induce sparsity onto the underlying atom graph, we achieve up to 2.26x less memory consumption and 2.04x less inference time depending on the atomic system and model, with little accuracy decrease across four models (MACE, Nequip, Orbv3, and TensorNet) – allowing the simulation and further study of significantly larger atomic systems at faster speeds.

## Acknowledgements

We would like to acknowledge Anthony Zhou for the intellectually stimulating conversations.

## Impact Statement

This paper presents work whose goal is to advance the field of Machine Learning. There are many potential societal consequences of our work, none which we feel must be specifically highlighted here.

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

## A. Proof of Differentiability

MLIPs capable of stable simulation typically perform message passing in the form of Equation 4, where an envelope function is applied to each message from all of the neighbors $u \in N_v$ of a center node $v$. The neighbors are computed to be all atoms within a fixed radius cutoff of $v$.

Message passing neural networks, where messages are constructed from the following form, are at least second-order differentiable.

$$M_{uv} = f\left(\frac{r_{uv}}{c_v}\right) m_{uv} \tag{4}$$

where $r_{uv}$ is the distance between $u$ and $v$, $c_v$ is the calculated cutoff radius for $v$, $m_{uv}$ is the rest of the message construction, and $f$ is a smooth function where $f(1) = 0$, $f'(1) = 0$, and $f''(1) = 0$.

When we take the gradient and Hessian of $M_{uv}$ with respect to the position of $v$, $\mathbf{x}_v$, we get the following

$$\nabla_{\mathbf{x}_v} M_{uv} = f'\left(\frac{r_{uv}}{c_v}\right) \nabla_{\mathbf{x}_v}\left(\frac{r_{uv}}{c_v}\right) m_{uv} + f\left(\frac{r_{uv}}{c_v}\right) \nabla_{\mathbf{x}_v} m_{uv} \tag{5}$$

$$
\begin{aligned}
H_{\mathbf{x}_v} M_{uv} = {} & f''\left(\frac{r_{uv}}{c_v}\right) \nabla_{\mathbf{x}_v}\left(\frac{r_{uv}}{c_v}\right)\left(\nabla_{\mathbf{x}_v}\left(\frac{r_{uv}}{c_v}\right)\right)^T m_{uv} + f'\left(\frac{r_{uv}}{c_v}\right) H_{\mathbf{x}_v}\left(\frac{r_{uv}}{c_v}\right) m_{uv} \\
& + f'\left(\frac{r_{uv}}{c_v}\right) \nabla_{\mathbf{x}_v}\left(\frac{r_{uv}}{c_v}\right)(\nabla_{\mathbf{x}_v} m_{uv})^T + f'\left(\frac{r_{uv}}{c_v}\right) \nabla_{\mathbf{x}_v} m_{uv}\left(\nabla_{\mathbf{x}_v}\left(\frac{r_{uv}}{c_v}\right)\right)^T + f\left(\frac{r_{uv}}{c_v}\right) H_{\mathbf{x}_v} m_{uv}
\end{aligned}
\tag{6}
$$

where

$$\nabla_{\mathbf{x}_v}\left(\frac{r_{uv}}{c_v}\right) = \frac{\nabla_{\mathbf{x}_v} r_{uv}}{c_v} - \frac{r_{uv} \nabla_{\mathbf{x}_v} c_v}{c_v^2}$$

$$H_{\mathbf{x}_v}\left(\frac{r_{uv}}{c_v}\right) = \frac{H_{\mathbf{x}_v} r_{uv}}{c_v} - \frac{\nabla_{\mathbf{x}_v} r_{uv}(\nabla_{\mathbf{x}_v} c_v)^T}{c_v^2} - \frac{\nabla_{\mathbf{x}_v} c_v(\nabla_{\mathbf{x}_v} r_{uv})^T + r_{uv} H_{\mathbf{x}_v} c_v}{c_v^2} + \frac{2 r_{uv} \nabla_{\mathbf{x}_v} c_v(\nabla_{\mathbf{x}_v} c_v)^T}{c_v^3}$$

The calculation for the gradient and second order gradients are similar for $\nabla_{\mathbf{x}_u} M_{uv}$ and $H_{\mathbf{x}_u}(M_{uv})$.

It is known that the composition of $k$-order differentiable equations is $k$-order differentiable. Therefore, as a result of Equations 5 and 6, and assuming $m_{uv}$ is second order differentiable, we simply need to prove that the calculation for $c_v$ is second-order differentiable with respect to the positions of $u$ and $v$.

### A.1. Cutoff Calculation

For convenience, we reiterate our dynamic cutoff function here:

Let $v$ be a node and $N_v$ be the set of incoming neighbors to $v$ within a hard cutoff radius of $h$. Let $u$ be another node such that $u \in N_v$, and let $r_{uv}$ describe the distance of the edge from atom $u$ to atom $v$.

For all $u \in N_v$, we define a rank $R_u$ where

$$R_u = \sum_{t \in N_v \setminus \{u\}} \left[ S(\alpha * (r_{uv} - r_{tv})) p\left(\frac{r_{tv}}{h}\right) \right] \tag{7}$$

where $S$ is the sigmoid function, $\alpha \in \mathbb{R}^+$, and $p : \mathbb{R} \to \mathbb{R}$ is the polynomial envelope function introduced in Gasteiger et al. (2020), defined as

$$
\begin{aligned}
p(x) = {} & 1 - \frac{(n+1)(n+2)}{2} x^n \\
& + n(n+2) x^{n+1} - \frac{n(n+1)}{2} x^{n+2}
\end{aligned}
$$

for some $n \in \mathbb{N}^+$ where $n \geq 3$. Note that, by design, $p(1) = 0$, $p'(1) = 0$, and $p''(1) = 0$. We perform the rankings in Equation 7 for all nodes $u$ in the atom graph $G$.

Next, we use the probability density function of the normal distribution to define a weighting function over neighbor rankings $R_u$. Let's define $\omega : \mathbb{R} \to \mathbb{R}$ as

$$\omega(x) = \frac{1}{\sigma \sqrt{2\pi}} e^{-\frac{1}{2}\left(\frac{x-\mu}{\sigma}\right)^2} \tag{8}$$

where $\mu, \sigma \in \mathbb{R}^+$. $\omega$ is interpreted as a symmetric weighting function for each $u \in N_v$ based on $u$'s ranking.

Finally, we calculate the dynamic cutoff, $c_v$ for node $v$ using a weighted average as follows:

$$c_v = f(N_v) = \frac{\left[\sum_{u \in N_v} \omega(R_u) p\left(\frac{r_{uv}}{h}\right) r_{uv}\right] + h\epsilon}{\left[\sum_{u \in N_v} \omega(R_u) p\left(\frac{r_{uv}}{h}\right)\right] + \epsilon} \tag{9}$$

where $\epsilon$ is a tiny value such as 1e-4 introduced to preserve numerical stability.

## A.2. Continuity and Differentiability of Dynamic Cutoff Function

**Theorem A.1.** *The cutoff calculation function always leads to second-order differentiable PES regardless of the movement of the atoms, including the addition and removal of atoms from the neighbor list.*

*Proof.* First, note that all operations in the cutoff calculation are smooth. Thus, for all $v \in V$, if every neighbor $u \in N$ is situated such that $r_{uv} < h$, then the cutoff calculation is smooth with respect to the positions of $u$ and $v$. Therefore, we only need to consider the case where there exists some neighbor $u$ such that $r_{uv} = h$.

Let $v \in V$. Assume that the number of atoms $s$ such that $r_{sv} < h$ is nonzero (otherwise, the would be no neighbor list for $v$). Assume that there exists some neighbor $t \in N$ such that $r_{tv} = h$. Then since $p(1) = 0$, $\omega(R_t) p\left(\frac{r_{tv}}{h}\right) = 0$. Therefore,

$$c_v = \frac{\sum_{u \in N} \omega(R_u) p\left(\frac{r_{uv}}{h}\right) r_{uv}}{\sum_{u \in N} \omega(R_u) p\left(\frac{r_{uv}}{h}\right)} = \frac{\sum_{u \in N \setminus \{t\}} \omega(R_u) p\left(\frac{r_{uv}}{h}\right) r_{uv}}{\sum_{u \in N \setminus \{t\}} \omega(R_u) p\left(\frac{r_{uv}}{h}\right)} \tag{10}$$

Therefore, if $t$ moves out of the $h$-radius sphere, its exclusion from $N$ will not cause a discontinuity in the value of $c$.

To define $\nabla_{\mathbf{x}_t} c_v$, we first define

$$A = \sum_{u \in N} \omega(R_u) p\left(\frac{r_{uv}}{h}\right) r_{uv}$$

$$B = \sum_{u \in N} \omega(R_u) p\left(\frac{r_{uv}}{h}\right)$$

Then

$$\nabla_{\mathbf{x}_t} A = \sum_{u \in N} \omega'(R_u) p\left(\frac{r_{uv}}{h}\right) r_{uv} \nabla_{\mathbf{x}_t} R_u + \left(\frac{1}{h} p'\left(\frac{r_{tv}}{h}\right) r_{tv} + p\left(\frac{r_{tv}}{h}\right)\right) \omega(R_t) \nabla_{\mathbf{x}_t} r_{tv}$$

$$\nabla_{\mathbf{x}_t} B = \sum_{u \in N} \omega'(R_u) p\left(\frac{r_{uv}}{h}\right) \nabla_{\mathbf{x}_t} R_u + \frac{1}{h} \omega(R_t) p'\left(\frac{r_{tv}}{h}\right) \nabla_{\mathbf{x}_t} r_{tv}$$

and

$$\nabla_{\mathbf{x}_t} c_v = \frac{\nabla_{\mathbf{x}_t} A}{B} - \frac{A \nabla_{\mathbf{x}_t} B}{B^2} \tag{11}$$

For all $u \in N \setminus \{t\}$,

$$\nabla_{\mathbf{x}_t} R_u = \left(-\alpha S'(\alpha(r_{uv} - r_{tv})) p\left(\frac{r_{tv}}{h}\right) + \frac{1}{h} S(\alpha(r_{uv} - r_{tv})) p'\left(\frac{r_{tv}}{h}\right)\right) \nabla_{\mathbf{x}_t} r_{tv}$$

Therefore, because $p(1) = 0$ and $p'(1) = 0$, $\nabla_{\mathbf{x}_t} R_u = 0$. Since every term concerning $t$ is also 0, it becomes clear that $\nabla_{\mathbf{x}_t} A = 0$ and $\nabla_{\mathbf{x}_t} B = 0$, so $\nabla_{\mathbf{x}_t} c_v = 0$. This implies that if $t$ moves away and is dropped from the neighbor list, this will not affect the gradient of the cutoff calculation with respect to the position of the atoms. Thus, $c_v$ is differentiable.

In addition,

$$
\begin{aligned}
H_{\mathbf{x}_t} A = &\sum_{u \in N} p\left(\frac{r_{uv}}{h}\right) r_{uv}(\omega''(R_u)(\nabla_{\mathbf{x}_t} R_u)(\nabla_{\mathbf{x}_t} R_u)^T + \omega'(R_u)H_{\mathbf{x}_t} R_u) \\
&+ \frac{1}{h}\left(\frac{1}{h}p''\left(\frac{r_{tv}}{h}\right) r_{tv} + 2p'\left(\frac{r_{tv}}{h}\right)\right) \omega(R_t)(\nabla_{\mathbf{x}_t} r_{tv})(\nabla_{\mathbf{x}_t} r_{tv})^T \\
&+ \left(\frac{1}{h}p'\left(\frac{r_{tv}}{h}\right) r_{tv} + p\left(\frac{r_{tv}}{h}\right)\right) (\omega'(R_t)(\nabla_{\mathbf{x}_t} r_{tv})(\nabla_{\mathbf{x}_t} R_t)^T + \omega(R_t)H_{\mathbf{x}_t} r_{tv})
\end{aligned}
$$

$$
\begin{aligned}
H_{\mathbf{x}_t} B = &\sum_{u \in N} p\left(\frac{r_{uv}}{h}\right) (\omega''(R_u)(\nabla_{\mathbf{x}_t} R_u)(\nabla_{\mathbf{x}_t} R_u)^T + \omega'(R_u)H_{\mathbf{x}_t} R_u) \\
&+ \frac{1}{h}\left(\omega'(R_t)p'\left(\frac{r_{tv}}{h}\right) (\nabla_{\mathbf{x}_t} r_{tv})(\nabla_{\mathbf{x}_t} R_t)^T + \frac{1}{h}\omega(R_t)p''\left(\frac{r_{tv}}{h}\right) (\nabla_{\mathbf{x}_t} r_{tv})(\nabla_{\mathbf{x}_t} r_{tv})^T + \omega(R_t)p\left(\frac{r_{tv}}{h}\right) H_{\mathbf{x}_v} r_{tv}\right)
\end{aligned}
$$

Therefore, $H_{\mathbf{x}_t} c_v$ is expressed as

$$
H_{\mathbf{x}_t} c_v = \frac{H_{\mathbf{x}_t} A}{B} - \frac{\nabla_{\mathbf{x}_t} A(\nabla_{\mathbf{x}_t} B)^T}{B^2} - \frac{\nabla_{\mathbf{x}_t} B(\nabla_{\mathbf{x}_t} A)^T + A H_{\mathbf{x}_t} B}{B} + \frac{2A\nabla_{\mathbf{x}_t} B(\nabla_{\mathbf{x}_t} B)^T}{B^3} \tag{12}
$$

Finally, notice that

$$
\begin{aligned}
H_{\mathbf{x}_t} R_u = &\left(\alpha^2 S''(\alpha(r_{uv} - r_{tv}))p\left(\frac{r_{tv}}{h}\right) - \frac{2\alpha}{h}S'(\alpha(r_{uv} - r_{tv}))p'\left(\frac{r_{tv}}{h}\right) + \frac{1}{h^2}S(\alpha(r_{uv} - r_{tv}))p''\left(\frac{r_{tv}}{h}\right)\right) (\nabla_{\mathbf{x}_t} r_{tv})(\nabla_{\mathbf{x}_t} r_{tv})^T \\
&+ \left(-\alpha S'(\alpha(r_{uv} - r_{tv}))p\left(\frac{r_{tv}}{h}\right) + \frac{1}{h}S(\alpha(r_{uv} - r_{tv}))p'\left(\frac{r_{tv}}{h}\right)\right) H_{\mathbf{x}_t} r_{tv}
\end{aligned}
$$

Substituting $p(1) = 0$, $p'(1) = 0$, and $p''(1) = 0$ reveals that $H_{\mathbf{x}_t} R_u = 0$ and that overall $H_{\mathbf{x}_t} A = 0$ and $H_{\mathbf{x}_t} B = 0$. Therefore, $H_{\mathbf{x}_t} c_v = 0$. Thus, if $t$ moves away from $v$ and is dropped from $N_v$, it will not affect $H_{\mathbf{x}_t} c_v$, so $c_v$ is always second-order differentiable.

These results imply that $c_v$ is second-order differentiable with respect to the positions of all atoms. Therefore, the message passing mechanism also remains second-order differentiable. $\square$

## B. Hyperparameter Details

For reproducibility, we share the hyperparameters used to train each of the models presented in this work. Note that, for comparisons between fixed cutoff and dynamic cutoff versions of the same model, we did not change any hyperparameters other than the cutoff type. Experiments were either performed using 2 H100-80GB GPUs or 1 A6000 GPU, depending on batch size requirements.

### B.1. MACE

Hyperparameter details found in Table 4.

### B.2. Nequip

Hyperparameter details found in Table 5.

### B.3. Orbv3

Orbv3 does not have an open source implementation to train the model from scratch. Rather, they have a barebones finetuning script. As a result, we wrote our own training script from scratch and open source it. We do not perform diffusion-based pretraining outlined in their original Orbv1 paper. Hyperparameter details found in Table 6.

| Category | Setting |
|---|---|
| Model type | ScaleShiftMACE |
| Number of interactions | 2 |
| Channels | 64 |
| $L_{\max}$ | 0 |
| Correlation order | 2 |
| Cutoff radius | 6.0 Å |
| Energy loss weight | 10 |
| Force loss weight | 1000 |
| SWA force weight | 10 |
| Batch size | 16 |
| Max epochs | 2000 |
| Early stopping patience | 15 |
| SWA | start @ 450 |
| Learning rate | 0.01 |

*Table 4.* MACE hyperparameters

| Category | Setting |
|---|---|
| Cutoff radius | 6.0 |
| Number of layers | 5 |
| $l_{max}$ | 1 |
| Number of features | 28 |
| Batch size | 32 |
| Max epochs | 1200 |
| Gradient clip | 1.0 |
| Energy weight | 1.0 |
| Force weight | 10.0 |
| LR Scheduler | Linear decay |
| Number of bessels | 8 |
| Learning rate | 0.01 |

*Table 5.* Nequip hyperparameters

| Category | Setting |
|---|---|
| Cutoff radius | 6.0 |
| Number of message passing steps | 5 |
| Latent Dimension | 128 |
| Base MLP hidden dimension | 64 |
| Base MLP depth | 2 |
| Head MLP hidden dimension | 64 |
| Head MLP depth | 2 |
| Batch size | 128 |
| Gradient clip | 1.0 |
| Warmup Epochs | 0.1% |
| Energy weight | 1.0 |
| Force weight | 1.0 |
| Max epochs | 1200 |
| Learning rate | 3e-3 |

*Table 6.* Orbv3 hyperparameters

### B.4. TensorNet

Hyperparameter details found in Table 7.

| Category | Setting |
|---|---|
| Cutoff radius | 6.0 |
| Number of message passing steps | 6 |
| Number of units | 64 |
| Batch size (MD22) | 2 |
| Batch size (MatPES) | 256 |
| Gradient clip | 2.0 |
| Energy weight | 1.0 |
| Force weight (MD22) | 1000.0 |
| Force weight (MatPES) | 1.0 |
| Max epochs | 2000 |
| Learning rate | 1e-3 |
| Early stopping patience | 30 epochs |

*Table 7.* TensorNet hyperparameters

## C. Melting Simulation Setup and Further Analysis

For the copper melting simulation, we initialized the system to contain 864 atoms of copper in an FCC configuration with a lattice constant of 3.61Å. We then used the Effective Medium Potential and Langevin thermostat with friction of 0.01 and a target temperature of 4500K to heat and melt the copper. In addition to Figure 7, we also plot a more comprehensive Figure 8 which includes more dynamic and fixed cutoff settings.

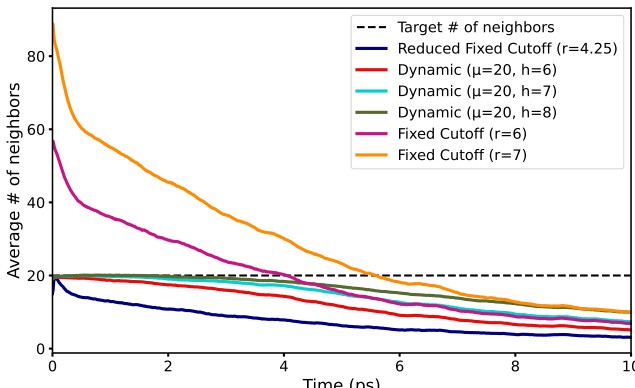

*Figure 8.* Similar to Figure 7, the average number of neighbors throughout a copper melting simulation at 4500K as a function of time. We plot varying fixed and dynamic cutoff settings. The reduced fixed cutoff of 4.25Å is chosen specifically to target around 20 neighbors in the bulk state of copper. The plot shows the inability of the reduced fixed cutoff to maintain a meaningful amount of neighbors relative to the unreduced cutoffs and dynamic cutoffs.

## D. Scaling number of layers

Depending on the model architecture, increasing the number of layers in the GNN-based MLIP increases the number of many-body interactions that can occur during energy calculation. The increased number of many-body interactions may be more desirable when there is an implicit target to the number of neighbors in an atomic system. We investigate this by training 6 TensorNet models with $\mu = 20$ of increasing layer count on the buckyball catcher dataset in MD22 (Simeon & De Fabritiis, 2023; Chmiela et al., 2023). We also fix the total number of parameters in each model to 455k parameters to isolate the effect of increasing layer count on downstream performance. We use the same training parameters and data

splits as used in Section 4.2. The results are presented in Figure 9. As seen from the figure, there is little noticeable effect of increasing layer count on either the force or energy errors.

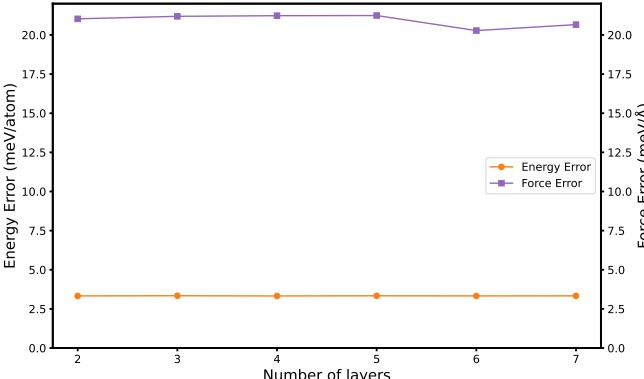

*Figure 9.* Scaling the number of layers in a dynamic cutoff TensorNet model (while keeping the total number of parameters fixed) with $\mu = 20$ on the buckyball catcher subset from the MD22 dataset. All else being equal, there is no noticeable accuracy difference between a dynamic cutoff model with a large amount of layers and a dynamic cutoff model with a smaller amount of layers.

## E. A Note on Neighbor Ranks

In Figure 10, for a random atom within a perturbed SiO2 supercell system consisting of 243 atoms, we show a parity plot between the dynamic ranks calculated in Equation 1 with respect to the true neighbor ranks. Notice how the dynamic ranks roughly follow the true neighbor ranks until the true neighbor rank becomes very large. This is a result of the $p(\frac{r_{tv}}{h})$ term within the summation. The term is maintains that, as the $t$ atom reaches the hard cutoff, the overall contribution of $t$ to the neighbor $u$ of atom $v$ approaches 0. Intuitively, this is to prevent the neighbor ranking of $u$ to change change if $t$ leaves the hard cutoff.

## F. Per-element Force MAE

We plot the relative increase in per-element force MAE between a dynamic cutoff model and its fixed cutoff counterpart in Figure 11. The model used were the TensorNet models trained in Section 4.3.

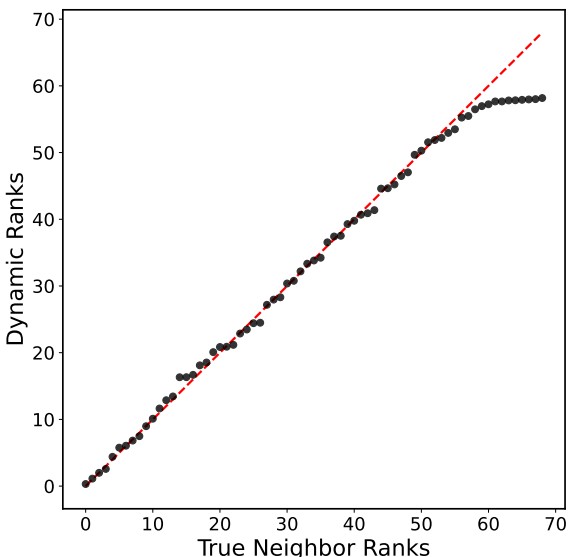

*Figure 10.* We show the dynamic ranks calculated by Equation 1 compared to the true neighbor ranks for the neighbors of a randomly selected atom in a 243 atom, randomly perturbed SiO2 supercell.

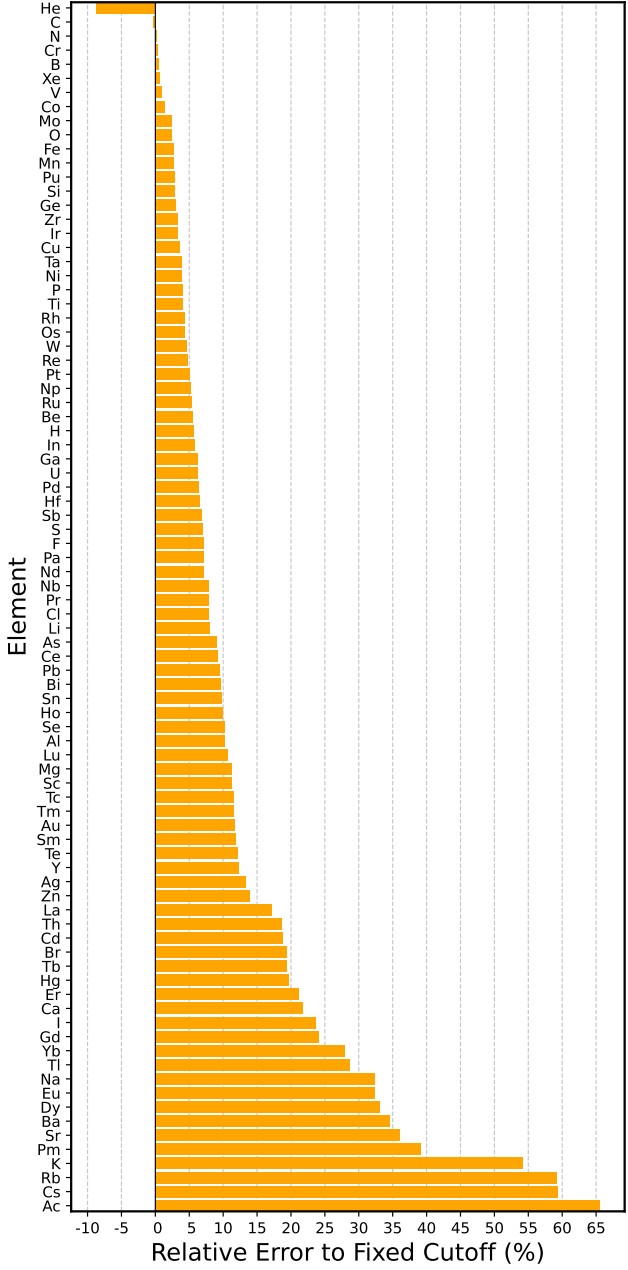

*Figure 11.* A element-by-element breakdown of the relative force error increase of the dynamic cutoff model compared to the fixed cutoff model baseline for the TensorNet ablation trained on the MatPES dataset (in section 4.3). The target number of neighbors, $\mu$, is 40. Elements with the largest atomic radii consistently show higher force error increases than smaller elements.

