# OpenReview forum: "Smooth Dynamic Cutoffs for Machine Learning Interatomic Potentials"
_ICML.cc/2026/Conference — ICML 2026 regular_

### Official Review · Reviewer_eJnS · 2026-02-19

**Soundness:** 3
**Presentation:** 3
**Significance:** 3
**Originality:** 2
**Overall Recommendation:** 5
**Confidence:** 4

**Summary:**

*New Comments post discussion:* After an extensive discussion period with the authors, I am happy to raise my original score of a 3 to the new score of a 5!

This paper seeks to make a portable framework for instituting dynamic cutoffs in MLIPs rather than using classic predefined cutoff radii. This cutoff focuses on number of neighbors for each node rather than an actual radius in angstroms. They show about a 2x reduction in memory and compute time. This does come at a cost in accuracy.

This is a very interesting direction of research. However, the concerns raised in Limitations 3-6 are currently preventing me from giving an accepting score. The results are only validated on unreasonably small models, despite the claim being an improvement in model scaling. Additionally, having a very non-negligible cost in accuracy to create a faster model is the opposite of what most researchers and practitioners are looking for. I look forward to an active discussion period with the authors about this work, as I think it has a lot of potential.

**Compliance With Llm Reviewing Policy:**

Affirmed.

**Final Justification:**

I believe that the authors have addressed all of my remaining concerns regarding the paper. They introduce a new method focused on efficiency in GNN-based machine learning force fields. I believe that it is an interesting technique that has shown sufficient success to be considered a useful contribution to literature. Runtime concerns are becoming increasingly important with the evolution of the field, motivating this work. Concerns about a lack of explanation have been well addressed in the rebuttal period; I know feel that the paper contains sufficient information to allow readers to adopt the technique with reasonable effort. Overall, I am happy to raise my initial score of a 3 to a 5.

Soundness: The technique is backed by experimental results showing that this technique improves efficiency while maintaining only a small loss in accuracy.

Originality: While the idea of controlling runtimes by adjusting cutoff radii is not unique, I have not seen this approach to it before. Other papers have looked at using non-trivial methods to create unique radii for each atom however. Overall, I think this is the weakest area of the submission since the fundamental idea has been approached before, just not in the proposed manner.

Significance: The experimental results justify that this technique gives a substantial enough performance gain to justify its use in some real-world scenarios. While certainly not applicable to every MLIP/MLFF use case, groups working on very large scale simulation or with constrained compute budgets could reasonably adopt this technique.

Clarity: Post revision, all my concerns about clarity have been addressed. Assuming that the authors' update the paper as they propose in the rebuttal, I do not find any issues that would likely trip up future readers.

**Key Questions For Authors:**

1 - What is a good energy drift value? Many machine learning researchers, even applied ones, may benefit from short reminders about technical terms.

2 - Why are alpha and n fixed to 10 and 50? This seems arbitrary. Are there experimental results that show that these are good choices? You say that you find $sigma$=4 to work well… can you show the experimental results that indicate this?

3 - You say: “one mathematical form of p was…” does this mean there is another form that you look at? Or is this just a way to say that your selection for p could be changed if a user wanted to do so?

4 - While I very much like Figure 2, it is presumably done for a single example. Is there a way that more examples or a numerical overview of this phenomena across a testing set could be provided?

5 - In the proof in Appendix A.1, is the epsilon included in the calculations during inference, or is it just included for convenience in the proof. If it is the second case (which I think it is), then the proof should conclude that it is approximately differentiable.

6 - In Section 4.5, you mention “cutting” MOF and protein systems to 1/3 their original size. How do you cut the systems? Can this be shown to not affect the problem? It would seem that removing part of the data could break the physical validity of the results. Is it a removal of examples in the data, or are the examples themselves shrunk?

7 - I am confused by the last sentence in Section 4.5: “Most computation time… number of edges”. Is this saying that the technique is slowing down the calculations, or is it just a comment on the fact that it induces a small overhead to construct neighbor lists?

**Limitations:**

1 - This proposed method does still include a hard cutoff hyperparameter.

2 - Studies on different graph types showing that the procedure works on different graph types would be very useful. Can it be shown to work on highly irregular graphs were nodes are heavily bunched in one area, but very spread out in another? What about a graph where all nodes are roughly equidistant from each other (doable with small slack distance epsilon such that all nodes are within distance [d-e, d+e] from each other). Section 4.5 would seem to indicate that this is not the case, and positive results rely on the common structure of atomistic graphs.

3 - The results are only shown on extremely small implementations of the models. Why is this? It raises concerns about the validity and benefit of the work when applied to full-scale systems. Especially in a situation where increasing scalability is the main contribution, showing performance on systems of scale is vital. The reduction of MOF and protein systems further raises concerns about the validity of those results.

4 - This technique is shown in Section 4.1 to reduce accuracy of the predictions. Modern ML is dominated by the need for more accuracy, even at the cost of much larger models and longer compute times. In addition, the compute time of MLIPs is almost always negligible when compared to that of DFT or other first-principles calculations, making an increase in model accuracy worth it even if it takes substantially longer than a smaller version. A 0.1-0.2 meV/atom gain in model force-prediction accuracy would be sufficient to publish a paper (maybe in a physical science rather than CS venue) in its own right.

5 - Another issue throughout the paper is the inclusion of statements about empirical results, without any studies or experiments to back them up.

6 - The statement that this could., in essence, reduce the adoption overhead by handling cutoff sizes from foundational models is interesting, but experimental results for it are not shown. It seems likely based on the given data, but is still only an assumption.

**Strengths And Weaknesses:**

**Strengths:**

1 - Very strong on novelty; I have not seen much research in the fundamentals of graph representations.

2 - The portability of the technique to work on many different MLIP architectures is impressive. Inclusion of four different state of the art models provides a strong argument that the results are generalizable.

3 - Covering both materials and molecular systems again shows generalizability of the results.

4 - The inclusion of gradient through the neighbor selection process is an oft overlooked ability, and its inclusion here is very nice to see.

5 - The inclusion of Theorem 3.1 is nice. I looked through the proof at a high level (since it is in the Appendix and not the main result of the paper I did not check all the arithmetic by hand), and I did have a question regarding it, which is included in Question 5. If the suppositions posed in Question 5 hold, then a small update should be made. Also, see Weakness 6.

6 - The argument at the end of Section 5.1 that this could reduce adoption is quite interesting. However, see Limitation 7.

**Weaknesses:**

1 - Materials simulations often do not require millions of MLIP calculations. They very often exist in the 40-50 frame range. Large biomolecules are the ones that can stretch to massive numbers of frames.

2 - In line 62-63 you mention it as weakness of Allegro to model local interactions. This is a weakness of almost all GNN-based architectures, not just Allegro.

3 - I find the 2x less memory to often be very confusing. Is it 1/2 or 1/3. Instead say: one-half reduction.

4 - Section 3 mentions that the use of w determines the average number of nodes you would like to have within the cutoff. While I like the inclusion of the function, in the earlier section it seems like there is a hard cutoff. While reading initially, I marked this down as a sever limitation of the paper until it later become apparent that the node count was not a single-fixed cutoff.

5 - The inability to enforce symmetry for inclusion in neighbor lists presents a concern about the “un-physicality” of this work. In a field where adherence to physical symmetries is prized, this raises a concern when considering adoption of this work. The paper mentions that this does not empirically lead to issues, but experimental results validating this are not shown.

6 - Even if the proof of Theorem 3.1 holds in theory, empirical evidence to back this up would be nice. One could conceive of a scenario in which the algorithm holds, but in practice its calculations result in compounding numeric errors.

7 - In Section 4.1 it once again gives seemingly arbitrary values for constants and says that they are supported by empirical tests, but the results of these tests are not shown.

---

> ### Author Rebuttal · Authors · 2026-03-29
>
> We thank you for the review! **We think there have been some misunderstandings which we hope to clear up but are pleased to hear that you enjoyed the approach overall!** Due to the 5000 character limit for rebuttals, we will try to tackle your biggest conceptual concerns. However, if we missed anything you feel is important, please feel free to bring them up again and we would be happy to address them in the next response stage.
>
> **Weaknesses**
> >W1
>
> Many molecular dynamics simulations of materials require nanoseconds (millions of timesteps) of simulation. We believe you may be confusing geometric relaxations (which only require 40-50 steps sometimes) with molecular dynamics simulations.
>
> >W2
>
> When we mentioned Allegro is limited to local, short-range interactions, we were referring to the limited 7 angstrom interaction radius of Allegro. The tiny 7A interaction radius has documented issues and results in poor performance in many cases compared to MLIPs with larger interaction radii. We will update the manuscript to be more clear.
>
> >W3
>
> We will update the manuscript to make this more clear.
>
> >W4
>
> That is correct, each atom has its own unique cutoff radius that dynamically modulates to target a pre-specified number of neighbors.
>
> >W5
>
> The symmetry of the neighborlist does not affect the physical symmetries (equivariance) embedded into the model. We will include a rigorous proof guaranteeing that the dynamic cutoff does not affect physical symmetries such as equivariance. In short, because the dynamic cutoff’s asymmetric graph topology remains invariant (pairwise distances are preserved), the aggregation of messages at each node maintains equivariance regardless of whether or not the edges themselves are bidirectional.
>
> >W6
>
> In Fig 3, we show the results of 4 300K DWNT NVE simulations and 4 3000K LiFePO4 NVE simulations with no drift or accumulation of numerical error. If there did exist compounding numeric errors, the energy drift would not be bounded and would instead explode during the simulation.
>
> **Questions**
> >Q1
>
> < 1meV/atom for 100 ps is the generally acceptable result for energy drift [1].
>
> [1] Fu, X., et al. (2025). Learning smooth and expressive interatomic potentials for physical property prediction. ICML.
>
> >Q2
>
> We find the dynamic cutoff is quite robust to different values of alpha, n, and sigma. We will expand Fig 10 in the appendix to show how the dynamic rankings change as these values are varied.
>
> >Q3
>
> One could also use the cosine envelope cutoff used in the TensorNet model.
>
> >Q4
>
> Please see our response to Reviewer F9HT’s W4/Q4 for more details on Fig 2! The purpose of Fig 2 was to show qualitatively the different “smoothness” of each approach as opposed to numerically study the phenomena.
>
> >Q5
>
> It is the first case - the epsilon is included in inference, making the dynamic cutoff provably differentiable both in practice and in theory.
>
> >Q6
>
> MACE, Orbv3, and Nequip all received the full system sizes. TensorNet ran into memory issues so we segmented the system into thirds, making sure to preserve the atomic densities (and thus the edge densities) so as to not affect relative timing or memory consumption results.
>
> >Q7
>
> It is a comment on the fact that it induces an extremely small fixed overhead to construct neighborlists. We will make this more clear.
>
> **Limitations**
> >L1
>
> The hard cutoff can be arbitrarily large and exists to set a maximum length on pairwise distances. Furthermore, the hard cutoff allows for retrofitting on highly-optimized neighborlist construction algorithms.
>
> >L2
>
> We run tests looking at the average number of neighbors of different atomic densities and report the values for a perturbed (eps=0.1) periodic SiO2 supercell (324 atoms):
>
> | Density (atoms/A^3) | Mean Neighbors | Std Neighbors |
> |-|-|-|
> |0.15| 20.19| 0.80|
> |0.13| 20.23| 0.79|
> |0.11| 20.27| 0.79|
> |0.09| 20.31| 0.79|
> |0.07| 20.36| 0.80|
> |0.05| 20.49| 0.80|
> The dynamic cutoff is highly robust to extreme changes in atomic density and perturbation.
>
> >L3
>
> The protein (1ADO) and MOF (H4Pb(C2O3)2) are full scale 16926 and 7500 atom systems respectively. We train on small implementations of the models since training >50 production sized models is computationally prohibitive. However, we also train a 25.1m parameter Orbv3 foundation model and show strong results. Please see our response to W2 from Reviewer 93ox.
>
> >L4
>
> We believe there is room in the field for slow, accurate ML models as well as fast, less accurate ML models. The dynamic cutoff advances the pareto frontier in simulation due to substantial increases in inference speed and memory consumption for only tiny accuracy reduction.
>
> >L5
>
> Please see our response to Q2 for how we addressed this.
>
> >L6
>
> The MatPES dataset we use for testing is a diverse foundational materials dataset for training foundational models. However, for results using a dynamic cutoff with a large, SOTA-level foundational model, please see our response to W2 from Reviewer 93ox.

---

> > ### Author Rebuttal · Reviewer_eJnS · 2026-03-31
> >
> > I thank the authors for their thorough response! The majority of my concerns have been addressed. Also, please do not feel the need to waste characters on formatting, greetings, etc. I understand the limited character constraint and am glad to forgo these if it helps the authors’ write a more complete response!
> >
> > **Unsatisfied Concerns:**
> >
> > **W7:**
> > In Section 4.1, you give values $\mu =40$ and $\mu=20$ for molecule and material systems respectively. Justification for these choices is that they are found to empirically work well. What other values did you try? Can you report results from them?
> >
> > **Q3:**
> >
> > From my understanding there could be infinitely many potential functions satisfying the key polynomial envelope constraint. Is this correct?
> >
> > **L1:**
> >
> > If the cutoff can be arbitrarily large, why include it? From my understanding, the algorithm selects nearest neighbors until the maximum number is reached, meaning the only point of the cutoff would be to restrict this selection if there are fewer neighbors than the listed target inside the threshold. Is it just for user preference that they would prefer interactions longer than a certain threshold not be included? Why would we want this?
> >
> > **L2:**
> >
> > I appreciate the construction of the results table. The robustness to the variation in atomic densities is impressive. I however, am still wondering if this addresses the concern about clustering of nodes. Does this address graphs which feature regions of tight bunches followed by sparse regions? I understand these are rare to observe in a physical sense, but it is instead a point of interest regarding the algorithm itself.
> >
> > **L4:**
> >
> > There is not too much more to say on this I believe: I agree that the tradeoff is potential useful. The decrease in accuracy observed on the new Orb experiments is not negligible and somewhat concerning, but is still small enough to make it a reasonable tradeoff.
> >
> > *The following concerns I believe to satisfied, provided the authors use the extra page or the Appendix to include the extra information. *
> >
> > **W1:**
> >
> > Yes, you are correct! I apologize for the confusion. Millions of frames is still on the large side even for MD, but it is not unreasonable.
> >
> > **W2:**
> >
> > Other common architectures like MACE, M3Gnet, CHGnet, etc default to 5-6 \AA, so the 7 \AA cutoff of Allegro is actually relatively high! Nonetheless, the point about local interactions is well taken… it is just not unique to Allegro.
> >
> > **W{3, 4, 5, 6}, Q{1, 2, 4, 5, 6, 7}, L{5, 6}:**
> >
> > Thanks for the updates! I have no further issues or comments with them.
> >
> > **L3:**
> >
> > I understand the main comparison is the model against itself with or without the dynamic cutoff radius. That being said. I would prefer to see a few training runs on production quality models rather than many runs on impractically small models. The inclusion of the Orb test satisfies this concern, and is, in my option, the strongest argument provided for the effectiveness of this technique.

---

> > > ### Author Response · Authors · 2026-04-02
> > >
> > > Thank you for the rebuttal acknowledgement, we’re glad we were able to address many of your concerns. Luckily we were able to fit a lot into that tiny 5000 character limit!
> > >
> > > We hope we will be able to address the rest of your concerns:
> > >
> > > >W7
> > > Our selection of \mu=20 for molecular systems comes from Fig 6., where we plotted force error with respect to varying choices of \mu. We then selected the \mu value proceeding a plateau in force error. Our selection of \mu=40 for materials systems comes from the same procedure except for the MatPES dataset. We trained TensorNet models of varying \mu values (20, 30, 40, 50, 60) on ~6% of the structures MatPES dataset. The structures were chosen to all share a set of 5 commonly occurring elements (hydrogen, oxygen, fluorine, phosphorus, sulfur). We then selected \mu=40 as the point proceeding a plateau in force error:
> > >
> > > | \mu  | Force MAE (meV/Å) |
> > > |----|----------------|
> > > | 20 | 71.1           |
> > > | 30 | 65.3           |
> > > | 40 | 62.9           |
> > > | 50 | 61.3           |
> > > | 60 | 60.0           |
> > >
> > > The resulting force values are significantly better than the training runs on the full MatPES dataset due to the overall training set and validation set being far smaller relative to the number of parameters in the TensorNet model. We chose the 5 elements on the basis of their commonality and frequency within the MatPES dataset. We used the same TensorNet architecture here as the rest of the paper and chose TensorNet specifically due to its stable and fast training dynamics.
> > >
> > > >Q3
> > > That is correct! The only constraints that the envelope function, p, must satisfy is for p(1) = 0, p′(1) = 0, and p′′(1) = 0. These are also the only assumptions we need regarding p in order to satisfy the theorem on smoothness.
> > >
> > > >L1
> > > For the construction and usage of the dynamic cutoff, we first consider all edges within a pre-selected hard cutoff, calculate the dynamic cutoff, and then prune the edges that are beyond the distance of the dynamic cutoff. Aside from this allowing us to retrofit to complex and highly optimized legacy neighborlist calculation code (primarily an engineering innovation), doing things in this way provides us the theoretical guarantees on smoothness due to the inclusion of the p(r_{uv}/h) term where h is the pre-selected hard cutoff.
> > >
> > > Intuitively, including edges beyond the nearest K neighbors during the construction of the dynamic cutoff makes sense for smoothness: as the K+1st neighbor approaches the Kth neighbor, we want the dynamic cutoff to approach the K+1st neighbor in a manner which counteracts the cusp created from the swapping of the Kth and K+1st neighbors. We can extend this intuition for the dynamic cutoff to also account for the K + 2nd neighbor, K + 3rd neighbor, and so on. In this extended intuition, the dynamic cutoff is adjusting its cutoff distance in the event that multiple neighbors simultaneously approach the Kth neighbor and initiate swaps with Kth neighbor.
> > >
> > > However, each node still has its own dynamic cutoff which modulates independently of the dynamic cutoffs of every other node. It’s only during the calculation of the dynamic cutoff do we first take the edges within some fixed cutoff. Afterwards, edges beyond the cutoff get pruned for the message passing phase, which is the most time and memory intensive phase. We will make this more clear in the manuscript.
> > >
> > > >L2
> > > You can consider a region of tight bunches as a high density system and then a sparse region as a low density system. With the table, we show that dynamic cutoff is extremely consistent across both regions of tight bunches and regions of sparseness. Therefore, we don’t expect there to be an impact of systems with wildly varying densities on the dynamic cutoff calculation. Our reasoning behind including the table was to show that, regardless of the density of any particular region or system, the dynamic cutoff remains very robust.
> > >
> > > >L4
> > > We would also like to point out that chemical accuracy for the OMol dataset is 1 kcal/mol which translates to ~43 meV/atom. The ~50% error increase in energies in the new Orbv3 experiments still only accounts for a small overall % of what is considered chemical accuracy.
> > >
> > > >W2
> > > One small correction: the cutoff radii for the construction of the underlying graph of these models are 5-6 angstroms. However, due to the message passing layers, the models can have interaction radii up to 60 or more angstroms [1].
> > >
> > > >W{1, 2, 3, 4, 5, 6}, Q{1, 2, 4, 5, 6, 7}, L{3, 5, 6}:
> > > We will include discussions and/or further elaborations on this by taking advantage of the extra page and appendix!
> > >
> > > We would like to thank you again for the fantastic discussion and we hope we were able to completely address your concerns! Thanks again!
> > >
> > > [1] https://arxiv.org/pdf/2506.23971

---

### Official Review · Reviewer_93ox · 2026-03-03

**Soundness:** 3
**Presentation:** 3
**Significance:** 2
**Originality:** 2
**Overall Recommendation:** 4
**Confidence:** 4

**Summary:**

This work aims to improve computational efficiency in machine-learning force fields by reducing the number of pairwise interactions considered within the graph neural networks. It does so by specifying a target number of neighbors for each atom. Given this target number and the spatial neighborhood of an atom, a per atom cutoff is derived such that in expectation the desired numbers of neighbors is met.

**Compliance With Llm Reviewing Policy:**

Affirmed.

**Final Justification:**

The authors provided sufficient data to convince me of the beneficial impact of their work. Though, it's final improvement on the accuracy/runtime tradeoff is somewhat limited. Thus, the weak accept.

**Key Questions For Authors:**

**Q1**: If the numbers of neighbors is fixed and does not vary significantly per atom, are rectangular tensor representations, e.g., N x K (for a maximum of K neighbors) more efficient? This might be relevant in models like UMA where a mixture of linear experts is used per edge.

**Q2**: For a fixed cutoff, radial basis functions encode specific distances. Now, if the cutoff changes, the basis functions encode different spatial distances. is this a problem for the MLPs that build on top? This could be measured e.g., if errors on atoms at periphery of a molecule have larger errors than those in the middle?

**Q3**: The absense of drift in Orbv3 is curious, given that previous literature found the absence of energy conservation in the model to cause energy drift? E.g., "Learning Smooth and Expressive Interatomic Potentials for Physical Property Prediction"

**Limitations:**

The authors do not discuss limitations of their method.

**Strengths And Weaknesses:**

## Strengths:

**S1**: The problem well identified and the solution via a smooth cutoff is a sensible approach.

**S2**: The resulting speedups are very welcome to the field.


## Weaknesses:

**W1**: The cutoff construction seems rather complicated with multiple hand-picked hyperparameters like $\alpha, \sigma, \mu$.

**W2**: In Table 1, one sees that newer and more expensive models generally perform better. It is questionable to use the older worse models for all ablations. Especially, since the gap widens for newer architectures compared to TensorNet. Given that the gap is larger the better the model, I wonder why smaller networks than the reference papers have been used. To make comparisons fair, it would be great to test with better models and normal sized models (at least in one comparison).

**W3**: The computational benefit is only measured in the number of neighbors. When accelerating a model, one could instead reduce hidden dimensions, etc. It would be good to see the trade-off between these actions and which one should do in pratice (smooth cutoff, reduce cutoff, reduce hidden dim, ...).

See questions.

---

> ### Author Rebuttal · Authors · 2026-03-29
>
> Thank you for the review! We’re pleased to see you enjoyed the sensibility of our approach and welcome the results. **However, there have been some significant misunderstandings that we hope to clear up!** Here are our point-by-point responses to your concerns:
>
> >W1: The cutoff construction seems rather complicated with multiple hand-picked hyperparameters like \alpha, \sigma, \mu.
>
> \mu refers to the target number of neighbors for each atom. Choosing \alpha directly corresponds to the sharpness of the smooth dynamic rankings calculated in Eq 1. We will bring Fig 10, which demonstrates this behavior, to the main body of the paper and further explain the use of \alpha. Finally, \sigma refers to the variance around \mu that you find acceptable. We will include more detailed elaboration on all of the values. However, we find that the dynamic cutoff behavior is rather robust to selection of \alpha and \sigma – we will include another figure showing how varying \alpha and \sigma values doesn't significantly affect downstream neighbor selection behavior. This figure will be similar to Fig 10.
>
> >W2: In Table 1, one sees that newer and more expensive models generally perform better. It is questionable to use the older worse models for all ablations. Especially, since the gap widens for newer architectures compared to TensorNet. Given that the gap is larger the better the model, I wonder why smaller networks than the reference papers have been used.
>
> We choose TensorNet as a test model due to its equivariant nature, simplicity, and fast training dynamics. Furthermore, TensorNet displays SOTA-level performance on the materials-based MatPES benchmark. **Our goal is not for performing comparisons across models. Rather, our goal is to compare fixed cutoff models to their direct dynamic cutoff counterparts.** Therefore, performance in Tab 1 and Tab 2 shouldn’t be compared between different model architectures.
>
> We also don’t perform experiments with production sized models (>~1milion parameters) due to the computational cost associated with training a large number of models for the experimental results we have shared.
>
> However, we also applied a dynamic cutoff to the 25.1 million parameter OrbMol model (which shows SOTA-level performance on the OMol25 dataset). After training the dynamic cutoff version of the model, we achieve the following results on the complete OMol25 validation set, showing that the results from the smaller models hold for larger model sizes.
>
> | Model | Energy Error (meV/atom) | Force Error (meV/Å) |
> |---|---|---|
> | Orbv3 OrbMol, fixed cutoff | 2.7 | 19.6 |
> | Orbv3 OrbMol, dynamic cutoff | 4.1 | 21.8 |
>
> >W3: The computational benefit is only measured in the number of neighbors. When accelerating a model, one could instead reduce hidden dimensions, etc. It would be good to see the trade-off between these actions and which one should do in pratice (smooth cutoff, reduce cutoff, reduce hidden dim, ...).
>
> **Accelerating MLIP inference through tuning hyperparameters such as the hidden dimension of the model are out of scope for this work.** We have an analysis on the detrimental effects of reducing the fixed cutoff compared to using a dynamic cutoff in Section 5.1.
>
> >Q1: If the numbers of neighbors is fixed and does not vary significantly per atom, are rectangular tensor representations, e.g., N x K (for a maximum of K neighbors) more efficient? This might be relevant in models like UMA...
>
> Under these assumptions, rectangular tensor representations should lead to speedup for UMA due to reduced irregular memory accesses on GPU. However, a thorough investigation on this is **out of scope** for our work.
>
> >Q2: For a fixed cutoff, radial basis functions encode specific distances. Now, if the cutoff changes, the basis functions encode different spatial distances. is this a problem for the MLPs that build on top?
>
> This is not a problem as the dynamic cutoff does not change distances between two atoms. The dynamic cutoff simply smoothly and continuously modulates the cutoff radius itself.
>
> >Q3: The absense of drift in Orbv3 is curious, given that previous literature found the absence of energy conservation in the model to cause energy drift? E.g., "Learning Smooth and Expressive Interatomic Potentials for Physical Property Prediction"
>
> The model displaying energy drift in “Learning Smooth and Expressive Interatomic Potentials for Physical Property Prediction” paper is the Orb direct force prediction model [1]. **This is a separate model from Orbv3, which is designed to be completely energy conserving [2].** Please refer to the Orbv3 paper for more details on its energy conservation behavior.
>
> [1] Fu, Xiang, et al. "Learning smooth and expressive interatomic potentials for physical property prediction." arXiv preprint arXiv:2502.12147 (2025).
>
> [2] Rhodes, Benjamin, et al. "Orb-v3: atomistic simulation at scale." arXiv preprint arXiv:2504.06231 (2025).
>
> Once again, we would like to thank you for the review!

---

> > ### Author Rebuttal · Reviewer_93ox · 2026-04-01
> >
> > I would like to thank the authors for their detailed rebuttal. I still have some follow up questions:
> >
> > W3: I disagree that such a comparison should be considered out of scope for a work promoting computational improvements. What is ultimately of interest is for a given compute budget what is the highest achievable accuracy.
> >
> > Q2: Is the dynamic cutoff only used within the neighborlist or also within the model, i.e., for rescaling the radial basis functions? If not, this would cause discontinuities in the PES when neighbors switch?

---

> > > ### Author Response · Authors · 2026-04-04
> > >
> > > Thank you for the rebuttal acknowledgement!
> > >
> > > >W3: I disagree that such a comparison should be considered out of scope for a work promoting computational improvements. What is ultimately of interest is for a given compute budget what is the highest achievable accuracy.
> > >
> > > This makes sense. We ran an additional experiment looking at whether to choose between a dynamic cutoff model with more parameters or a fixed cutoff model with less parameters. TensorNet was chosen due to its fast training dynamics, SOTA-level performance on the MatPES dataset, and simple hyperparameters.
> > >
> > > We train a TensorNet model with 32 hidden units and a fixed cutoff radius of 6 angstroms on the MatPES dataset. In order to choose the number of hidden units to be 32, we measured the inference time and memory consumption on supercells of the MatPES validation set across a sweep of hidden unit values:
> > >
> > > | units | mean_ms | std_ms | peak_mem_mb |
> > > |-------|--------:|-------:|------------:|
> > > | 8     | 152.2   | 8.3    | 7753.7      |
> > > | 16    | 164.8   | 20.7   | 15155.8     |
> > > | 24    | 200.6   | 8.7    | 22502.0     |
> > > | 32    | 256.6   | 2.0    | 29899.2     |
> > > | 34    | 273.1   | 10.9   | 31760.9     |
> > > | 36    | 282.8   | 16.5   | 33653.1     |
> > > | 38    | 287.8   | 20.9   | 35396.1     |
> > > | 40    | 299.0   | 16.1   |  37252.2 |
> > >
> > > We then selected 32 hidden units as it most closely matched the inference time and memory consumption profile of the dynamic cutoff model targeting 40 neighbors:
> > >
> > > | units | mean_ms | std_ms | peak_mem_mb |
> > > |-------|--------:|-------:|------------:|
> > > | 64    | 269.3   | 11.8   | 27502       |
> > >
> > > We then trained the 32 hidden unit, 6 angstrom fixed cutoff model and achieved the following results. We also include results from Table 3 for your convenience where we select a reduced fixed cutoff to average 40 neighbors on the MatPES dataset (5.4141 angstroms).
> > >
> > > | Method                         | Units | Energy MAE | Force MAE |
> > > |--------------------------------|-------|------------|-----------|
> > > | Fixed cutoff (cutoff = 6)      | **32**    | 52         | 175       |
> > > | Fixed cutoff (cutoff = 5.4141) | 64    | 53         | 171       |
> > > | Dynamic cutoff (μ = 40)       | 64    | **43**         | **160**       |
> > >
> > >
> > > This shows that, **given fixed amortized inference compute, training dynamic cutoff models significantly outperform their non-dynamic cutoff model counterparts.** We would like to thank the reviewer for pushing us to perform this additional experiment. You made a great insight that has helped us further our work!
> > >
> > >
> > > >Q2: Is the dynamic cutoff only used within the neighborlist or also within the model, i.e., for rescaling the radial basis functions? If not, this would cause discontinuities in the PES when neighbors switch?
> > >
> > > The dynamic cutoff is used within the model as well! Radial basis functions are provided with both the dynamic cutoff radius as well as the pairwise distance. Because the dynamic cutoff moves in a smooth manner, the ratio between pairwise distance and dynamic cutoff in radial basis functions such as Bessel functions also move in a smooth manner. This maintains the overall smoothness of the PES.
> > >
> > > We apologize we weren’t able to elaborate more deeply on this earlier due to the character limit.
> > >
> > > We would like to thank you for the thoughtful reviews and hope we were able to completely address your concerns!

---

### Official Review · Reviewer_F9HT · 2026-03-05

**Soundness:** 3
**Presentation:** 2
**Significance:** 3
**Originality:** 3
**Overall Recommendation:** 3
**Confidence:** 3

**Summary:**

The paper proposes a framework for constructing a dynamic cutoff function for machine learning interatomic potentials used in molecular dynamics simulations. Instead of employing a fixed cutoff radius for all atoms in the molecular graph, the method introduces a ranking-based mechanism that determines an individualized cutoff for each atom. The authors show that the resulting force field remains twice differentiable. Empirically, the dynamic cutoff induces sparsity in the interaction graph while maintaining nearly the same MAE as fixed-cutoff models. In practice, the resulting graphs exhibit sparsity comparable to those produced by $k$-nearest-neighbor constructions, leading to improved memory efficiency and faster inference during simulation.

**Compliance With Llm Reviewing Policy:**

Affirmed.

**Final Justification:**

The paper addresses an important issue in atomistic machine learning and proposes a promising dynamic cutoff mechanism to replace hard neighbor selection with a smoother, differentiable alternative, my final evaluation is **weak reject**. I appreciate the authors’ thoughtful and detailed rebuttal, which clarified several of my follow-up questions, including the use of autograd-derived forces, the applicability to periodic systems, and the role of removed edges in long-range interactions. These responses improved my understanding of the method and increased my confidence that some of the concerns can be addressed in a revision. But in my point of view, the presentation of the paper, especially regarding the illustration of the methods, and study of the ablated interaction should be more emphasized. I feel the current manuscripts fall short in that sense.

**Key Questions For Authors:**

1. Can the authors include a **schematic illustrating how the dynamic cutoff graph is constructed** to clarify Eq. 1, which currently appears abrupt and lacks visual explanation?

2.
   Can the authors **overlay results from fixed-cutoff and KNN graph constructions** in Figure 3 (energy drift) and include **fixed-cutoff models with varying cutoff radii and KNN models with varying (K)** in Figure 6 to better compare sparsity–accuracy trade-offs?

3.
   Can KNN graphs **achieve comparable MAE** to the dynamic cutoff method, and can the authors provide **a deeper analysis comparing KNN and dynamic cutoff graphs**, including structural similarities (neighbor counts) and differences in model behavior?

4.
   What exactly is the **“random system” used in Figure 2**, how is it constructed, and is the **jagged PES intrinsic to KNN or sensitive to the choice of (K)** or local atomic density?

5.
   Does the **MatPES-r2SCAN dataset include periodic boundary conditions**, and if so, how does the proposed dynamic cutoff formulation **handle or extend to periodic systems**?

6.
   How does **reducing the number of edges affect the modeling of long-range interactions**, and can the authors provide an **analysis of the removed edges** (e.g., distribution by interatomic distance or atom type) to show that pruned interactions are less physically important?

**Limitations:**

yes

**Strengths And Weaknesses:**

Strengths:
* The submission is supported by an extensive experimental evaluation across multiple models and datasets. I appreciate the breadth of the empirical study, which provides convincing evidence of the practical utility of the proposed dynamic cutoff mechanism. The results consistently demonstrate improved efficiency while largely preserving predictive accuracy, suggesting that the approach is a useful engineering improvement for large-scale MLIP simulations.

Weaknesses:
* The presentation of the paper could be improved: the authors should consider including a schematic illustrating the construction of the dynamic cutoff graph, which would help clarify the method. As it stands, Eq.1 reads as somewhat abrupt and lacks visual support.
* In Figure 3, the paper only reports the energy drift for the dynamic cutoff method. For a clearer comparison, the authors should also overlay the results from the fixed-cutoff and $k$-nearest-neighbor (KNN) graph constructions. Similarly, in Figure 6 it would be helpful to include the performance of fixed-cutoff models on the two molecules under varying cutoff radii and the performance of KNN with varying K. Such comparisons would make it easier to understand the trade-off between sparsity and accuracy across different graph construction strategies.
* In Figure 2, the paper shows that KNN graphs can lead to jagged energy surfaces, but it remains unclear whether KNN graphs could achieve comparable MAE to the dynamic cutoff method across the evaluated datasets. Since the dynamic cutoff graphs appear structurally similar to KNN graphs (both yielding an approximately fixed number of neighbors per atom), it would be useful to provide a more detailed analysis of their similarities and differences, both structurally and in terms of model behavior.
* I remain suspicious on the generalizability of Figure 2. It would also be helpful to clarify the setup used in Figure 2. The paper states that the example corresponds to a “random system,” but the exact system configuration is not specified. Providing more details about how this system is constructed would help with the generalizability of the phenomenon. In addition, it would be useful to discuss whether the observed jagged potential energy surface (PES) is intrinsic to the KNN construction or sensitive to the choice of $K$. For example, I expect the the PES become smoother if a larger $K$ were used, or if the density of neighboring atoms changed?
* It is not clear whether the MatPES-r2SCAN dataset used in the experiments includes periodic boundary conditions. If periodic boundary conditions are present, the authors should clarify any associated caveats. It would also strengthen the paper to discuss how the proposed dynamic cutoff formulation could be extended to periodic systems.
* Reducing the number of edges may limit the model’s ability to capture longer-range interactions. It would therefore be helpful if the authors provided additional analysis of how the removal of edges affects the modeling of long-range interactions. For example, an analysis of the eliminated edges, such as the distribution of removed interactions by interatomic distance or atom type. This would help clarify whether the pruned interactions are indeed physically less important.

---

> ### Author Rebuttal · Authors · 2026-03-31
>
> We would like to thank the reviewer for the very detailed and sharp review! We’re glad to hear that the weaknesses are primarily related to the presentation of the work as opposed to the novelty or strength of the work itself. We will include all of the points raised in your review directly into the paper.
>
> >W1/Q1
>
> We have Figure 10 in the appendix outlining a comparison between the dynamic smooth ranks calculated in Eq. 1 and the true neighbor ranks. We will bring this figure to the main body along with further explanation of the “sum of sigmoids” approach and another figure outlining an example scenario with example u nodes, v nodes, and t nodes.
>
> >W2/Q2
>
> Fig 3: We trained 8 KNN models (4 model architectures with k=20 for DWNT and k=40 for LiFePO4) on DWNT and MatPES-r2scan. After running the same simulation setup for DWNT and high temperature LiFePO4, we find the energy drift to explode almost immediately for both systems - indicating highly unusable and unphysical simulation. We will include these results into Fig 3.
>
> Fig 6: For both DWNT and buckyball systems, we trained more fixed cutoff models at 3, 4, and 5 angstroms. We also trained KNN models with k=5, 10, 15, 20, 30, 40. Below are the force error (meV/A) results that we will include in the plot:
>
> Buckyball
> | Method   | Value (meV/Å) |
> |--|--|
> | K=5| 128.95|
> | K=10| 37.50|
> | K=15| 32.52|
> | K=20| 28.11|
> | K=30| 23.89|
> | K=40| 22.96|
> | fixed=3| 29.05|
> | fixed=4| 22.24|
> | fixed=5| 21.42|
>
> DWNT
> | Method   | Value (meV/Å) |
> |--|--|
> | K=5| 245.34|
> | K=10| 102.71|
> | K=15| 67.40|
> | K=20| 56.87|
> | K=30| 47.59|
> | K=40| 23.08|
> | fixed=3| 57.24|
> | fixed=4| 39.36|
> | fixed=5| 33.75|
>
> The dynamic cutoff models consistently outperform their corresponding KNN models due to the additional inductive bias/generalizability associated with the smooth energy surfaces.
>
> >W3/Q3
>
> Regarding MAE performance:
> We train a KNN TensorNet model on the MatPES dataset with k=40 to match the dynamic cutoff \mu=40. The model achieves 54 meV/atom energy error and 174 meV/A force error. Dynamic cutoff models outperforming KNN models makes sense due to the additional generalizability associated with smooth energy and force predictions on hidden test data.
>
> Regarding analysis:
> We performed an analysis studying the behavior of the KNN graph (k=40) model trained on MatPES on atoms with varying atomic radii similar to Fig 5, testing for how the model behaves when slightly longer ranged interactions are required. We find no relationship between the atomic radii length and the relative error increased compared to baseline. This makes sense as the KNN graph does not suffer from the same neighbor selection dropout issues that the dynamic cutoff suffers from (described in Sec 4.7).
>
> For Fig 7, we will also include a curve showing the neighbor inclusions of KNN graphs, allowing for a direct comparison between the neighbor count behavior of the dynamic cutoff graphs compared to the KNN graphs at varying atomic densities.
>
> >W4/Q4
>
> The “random system” was a 0.2A randomly perturbed SiO2 cell with pbc enabled. We will modify the manuscript to be more clear on this. In order to calculate the surface, we translated a randomly selected atom from a (5, 5, 5) supercell along the xy plane and calculated the energy at each point along the plane. In order to check the generalizability of the jagged phenomenon, we also performed the same qualitative smoothness tests on 3 other systems using trained TensorNet models on MatPES: H2O, LiFePO4, and MgO. For the max neighbors (KNN) tests, we used K= 5, 10, 20, 30, 40 and found the same jagged behavior regardless of K. The qualitative behavior doesn’t significantly change due to the jagged phenomena being a function of undifferentiable neighbor switches as the atom translates along the xy plane – each neighbor switch leads to a “jag”/undifferentiable point in the surface. The number of neighbor switches is more a function of the atomic density and less so a function of the value of K.
>
> The fixed cutoff and dynamic cutoff surfaces were all qualitatively smooth. We will include all of the additional surfaces in the appendix as well as details on experiment setup.
>
> >W5/Q5
>
> The MatPES dataset uses full periodic boundary conditions. Dynamic cutoffs can be natively applied to periodic systems without any adjustment. The provable differentiable guarantees still apply as there are no assumptions made on the periodicity of the atomic system or the effects that periodicity has on the underlying graph.
>
> >W6/Q6
>
> Here is the distribution of lengths for edges that are maintained after pruning (\mu=40) across the MatPES dataset (sorted by removed length). We will include the entire table and full analysis in the paper.
>
> | Element | Avg removed edge length (Å) | # removed edges |
> |---------|------------------------------|------------------|
> | N| 5.2617| 7,135,280|
> | B| 5.3399| 2,490,184|
> | Li| 5.3791| 1,651,141|
> | ...| ...| ...|
> | Se| 5.7379| 277,239|
> | Xe| 5.7386| 4,312|

---

> > ### Author Rebuttal · Reviewer_F9HT · 2026-04-01
> >
> > W1-Q1. I think this is not enough. Need a schematics to make the dynamic ranking procedure more intuitive rather than relying on the equations. Right now, it is very hard to parse.  But i think the author could address this easily.
> > W2-Q2. I have a follow-up question: which mode you used to derive the forces (autograd/or direct regression) in these 8 architectures? I am asking because another source of energy drift could be coming from a non-conservative force field.
> > W5/Q5. How? Can the author detail the conversion more? I think this will improve the clarity of the paper.
> > W6/Q6. Are the removed edges important for long-range interaction?

---

> > > ### Author Response · Authors · 2026-04-01
> > >
> > > Thanks for the rebuttal acknowledgement!
> > >
> > > >W1-Q1. I think this is not enough. Need a schematics to make the dynamic ranking procedure more intuitive rather than relying on the equations. Right now, it is very hard to parse. But i think the author could address this easily.
> > >
> > > Understood, we will address this by creating a figure with a center atom, multiple neighbors, as well as arrows depicting how dynamic ranking changes as those neighbors move.
> > >
> > > >W2-Q2. I have a follow-up question: which mode you used to derive the forces (autograd/or direct regression) in these 8 architectures? I am asking because another source of energy drift could be coming from a non-conservative force field.
> > >
> > > For the 8 models, we calculated forces using the autograd mode.
> > >
> > > >W5/Q5. How? Can the author detail the conversion more? I think this will improve the clarity of the paper.
> > >
> > > Of course! Periodic systems introduce two potential phenomena that don't exist in non-periodic systems: 1) there may exist multiple edges from node A to node B, and 2) there may exist nodes in the graph that have a non-zero amount of self-edges. The key insight into why the dynamic cutoff works natively for periodic systems comes from the observation that the dynamic cutoff only incorporates information regarding incoming edge length and *not the identity or location of the source node for the incoming edge*. Therefore, it doesn't matter for the calculation of the dynamic cutoff if there exist multiple edges from node A to node B or if there exist self edges within the graph. The implementation still holds as does the theoretical guarantees that the dynamic cutoff offers. We will make this more clear in the manuscript.
> > >
> > > >W6/Q6. Are the removed edges important for long-range interaction?
> > >
> > > We attempt to study how important the removed edges are for longer-ranged interactions in Sec 4.6 and Fig 5. We showed that atoms with larger atomic radii (e.g. K, Ac, Rb, Cs) result in disproportionately higher error increases relative to the fixed cutoff baseline which incorporates all of the edges. In contexts where longer-ranged interaction is not as important, such as atoms with smaller atomic radii (e.g. N, C, He), we find that the removed edges have no effect. On these elements, the dynamic cutoff model (removing furthest edges) performs similarly or even outperforms the fixed cutoff models (removing no edges). We will elaborate on this and make this more clear in the manuscript.
> > >
> > > We hope we were able to thoroughly answer your questions and want to thank you for making this overall review process smooth!

---

### Official Review · Reviewer_1y1L · 2026-03-10

**Soundness:** 2
**Presentation:** 3
**Significance:** 2
**Originality:** 3
**Overall Recommendation:** 3
**Confidence:** 4

**Summary:**

This paper proposes a smooth dynamic cutoff scheme for MLIPs. The main idea is to replace the standard fixed cutoff with a per-atom adaptive cutoff while keeping the potential energy surface smooth. The method uses a soft ranking of neighbors, a smooth weighting function, and a weighted average to compute the cutoff. The authors implement the method in four MLIPs, including MACE, NequIP, Orbv3, and TensorNet. Experiments on MD22 and MatPES show lower memory usage and faster inference, with moderate accuracy degradation compared with fixed cutoffs.

**Compliance With Llm Reviewing Policy:**

Affirmed.

**Final Justification:**

This manuscript attempts to assess the central issue of whether one can induce sparsity in MLIP atom graphs through a dynamic, per-atom cutoff while still preserving PES smoothness and simulation stability. The paper makes a meaningful technical case through a differentiability argument and empirical evidence, including 100 ps NVE tests and reported gains of up to 2.26x lower memory use and 2.04x faster inference.

I thank the authors for the detailed rebuttal. The rebuttal addressed most of my concrete concerns by providing finer-grained inference-time breakdowns, training time, peak-memory tables, and relative-error comparisons, which improved the clarity of the work and made the source of the speedup more transparent. While my main concern is that the relative error increases are not always negligible for some settings, even when the absolute degradation is small.

My final recommendation is therefore unchanged: I view the paper as interesting and potentially useful, and the rebuttal improved my confidence in the presentation, but it did not fully change my caution that the adaptive cutoff remains a lossy approximation whose practical adoption may be limited in applications that demand very high MLIP accuracy and reliability.

**Key Questions For Authors:**

Please refer to Weakness.

**Limitations:**

Yes

**Strengths And Weaknesses:**

Strengths: The proposed method is well motivated. Most existing efforts to accelerate MLIP inference focus on kernel optimization, system-level implementation, and so on. This paper explores a different method by dynamically adjusting the neighbor set.

Weaknesses: 1) The efficiency analysis. In Figure 5, we encourage the authors to provide a finer-grained time breakdown. Under the fixed-cutoff setting, please decompose the absolute runtime into graph construction, forward energy prediction, backward force prediction, and other components. Under the dynamic cutoff setting, please separate into graph construction, neighbor preprocessing overhead, forward energy prediction, backward force prediction, and others. This would make the source of the speedup much clearer.
2) The analysis of model training. Please report the convergence curves, end-to-end training time, and training memory reduction.
3) The training errors. The absolute error increase is often small, but the relative increase is sometimes not small, especially in Table 1.

---

> ### Author Rebuttal · Authors · 2026-03-29
>
> We are pleased to hear that your primary criticism of our work lies in reporting more values (e.g. reporting inference and training breakdown) as opposed to novelty, impact, or motivation! We would also like to thank you for the review!
>
> Below are our point by point updates to how we reported values in the work:
>
> >1) The efficiency analysis. In Figure 5, we encourage the authors to provide a finer-grained time breakdown. ... This would make the source of the speedup much clearer.
>
> Here are the breakdowns of inference times used in Fig 5. “Fwd” refers to the force energy prediction. “Bwd” refers to backward force prediction.
>
> TensorNet
> | System | Mode   | Fwd (%) | Bwd (%) | Dynamic Cutoff (%) |
> |--------|--------|--------:|--------:|----------:|
> | 1ADO   | fixed  | 21.4%   | 78.6%   | —         |
> | 1ADO   | dyn_20 | 20.0%   | 79.8%   | 0.2%      |
> | H4PB   | fixed  | 23.3%   | 76.7%   | —         |
> | H4PB   | dyn_40 | 23.9%   | 75.8%   | 0.3%      |
>
> MACE
> | System | Mode   | Fwd (%) | Bwd (%) | Dynamic Cutoff  (%) |
> |--------|--------|--------:|--------:|----------:|
> | 1ADO   | fixed  | 26.2%   | 73.8%   | —         |
> | 1ADO   | dyn_20 | 29.8%   | 67.6%   | 0.9%      |
> | H4PB   | fixed  | 25.2%   | 74.8%   | —         |
> | H4PB   | dyn_40 | 31.9%   | 67.3%   | 0.8%      |
>
> Nequip
> | System | Mode   | Fwd (%) | Bwd (%) | Dynamic Cutoff  (%) |
> |--------|--------|--------:|--------:|----------:|
> | 1ADO   | fixed  | 30.2%   | 69.8%   | —         |
> | 1ADO   | dyn_20 | 32.6%   | 67.0%   | 0.4%      |
> | H4PB   | fixed  | 30.1%   | 69.9%   | —         |
> | H4PB   | dyn_40 | 31.8%   | 67.8%   | 0.4%      |
>
> Orbv3
> | System | Mode   | Fwd (%) | Bwd (%) | Dynamic Cutoff (%) |
> |--------|--------|--------:|--------:|----------:|
> | 1ADO   | fixed  | 34.1%   | 65.9%   | —         |
> | 1ADO   | dyn_20 | 33.5%   | 66.2%   | 0.3%      |
> | H4PB   | fixed  | 32.1%   | 67.9%   | —         |
> | H4PB   | dyn_40 | 30.2%   | 69.5%   | 0.3%      |
>
> >2) Please report the convergence curves, end-to-end training time, and training memory reduction.
>
> Here are tables of end to end training time and maximum memory allocation during the training runs reported in Table 1 and Table 2. The inference speed increase of the dynamic cutoff models is countered by slower convergence during training.
>
> MatPES dataset, fixed cutoff
> | Model      | Train Time (hrs) | Peak Memory (GB) |
> |------------|-----------------|-----------|
> | Orb        | 241.3           | 15.2      |
> | TensorNet  | 23.8            | 70.3   |
> | Nequip     | 48.8            | 14.9      |
> | MACE       | 189.5           | 8.6       |
>
> MatPES dataset, dynamic cutoff
> | Model      | Train Time (hrs) | Peak Memory (GB) |
> |------------|-----------------|-----------|
> | Orb        | 240.0           | 6.6       |
> | TensorNet  | 24.1            | 56     |
> | Nequip     | 49.0            | 5.3       |
> | MACE       | 191.0           | 4.4       |
>
> MD22 dataset, fixed cutoff:
>
> | Model     | Avg Training Time (hrs) | Avg Peak Memory (GB) |
> |-----------|------------------------|----------------------|
> | Orb       | 32.1                   | 12.20               |
> | TensorNet | 7.17                   | 32.99               |
> | Nequip    | 10.81                  | 27.69               |
> | MACE      | 15.50                  | 11.84               |
>
> MD22 dataset, dynamic cutoff:
> | Model     | Avg Training Time (hrs) | Avg Peak Memory (GB) |
> |-----------|------------------------|----------------------|
> | Orb       | 37.56                  | 7.76                |
> | TensorNet | 6.97                   | 19.83               |
> | Nequip    | 10.21                  | 17.86               |
> | MACE      | 17.37                  | 7.01                |
>
> >3) The relative training errors.
>
> Here are tables on the relative error increases for the MD22 and MatPES models:
>
> MD22 averaged relative error of dynamic cutoff vs. fixed cutoff:
>
> | Model      | Energy Δ (%) | Force Δ (%) |
> |------------|-------------|-------------|
> | MACE       | +6.5%       | +11.3%      |
> | Nequip     | +33.1%      | +7.0%       |
> | Orbv3      | +5.2%       | +8.6%       |
> | TensorNet  | -0.4%       | +6.4%       |
>
> MatPES relative error of dynamic cutoff vs. fixed cutoff:
> | Model      | Energy Δ (%) | Force Δ (%) |
> |------------|-------------|-------------|
> | MACE       | +3.3%       | +9.8%       |
> | Nequip     | +18.6%      | +0.6%       |
> | Orbv3      | +1.5%       | +0.6%       |
> | TensorNet  | +10.3%      | +5.3%       |
>
> As molecular dynamics practitioners ourselves, we would like to highlight that the absolute error increases of 0.1-0.2 meV/atom for energies are incredibly trivial from a practical sense.

---

> > ### Author Rebuttal · Reviewer_1y1L · 2026-04-02
> >
> > The authors said: “As molecular dynamics practitioners ourselves, we would like to highlight that the absolute error increases of 0.1–0.2 meV/atom for energies are incredibly trivial from a practical sense.” I have three follow-up questions:
> >
> > 1) When designing a smooth cutoff method, I think the authors should try to preserve consistency with the original method as much as possible in terms of accuracy. The authors argue that an ABE increase of 0.1–0.2 meV/atom is negligible in practice. I wonder: if Rc were reduced further, how much additional loss may occur in this setting?
> >
> > 2) MD22 is a relatively simple dataset. I would suggest evaluating the method on more challenging datasets. For example, it would be helpful to test ORB and MACE on OMol.
> >
> > 3) For some MLIPs that involve bond graphs, the construction of the bond graph and feature updates can be more memory-intensive. I suggest that the authors evaluate CHGNet and DPA3, and report both the accuracy and the memory reduction by using the proposed method in the paper.

---

> > > ### Author Response · Authors · 2026-04-02
> > >
> > > Thank you for the acknowledgment!
> > >
> > > >When designing a smooth cutoff method, I think the authors should try to preserve consistency with the original method as much as possible in terms of accuracy. The authors argue that an ABE increase of 0.1–0.2 meV/atom is negligible in practice. I wonder: if Rc were reduced further, how much additional loss may occur in this setting?
> > >
> > > We ran this experiment and provided analysis in Section 5.1/Table 3 of the manuscript. We compared a dynamic cutoff model with a target of 40 neighbors with a reduced fixed cutoff model where the reduced cutoff is chosen to average 40 neighbors for the dataset. We trained these two models on the challenging foundational MatPES dataset [1] . We have copied Table 3 here for your convenience:
> > >
> > > | Cutoff Type                         | Energy | Forces |
> > > |-------------------------------------|--------|--------|
> > > | Dynamic Cutoff                      | 43     | 160    |
> > > | Reduced Fixed Cutoff (5.4141 Å)     | 53     | 171    |
> > >
> > > This shows that simply reducing the cutoff Rc to a specific neighbor average (i.e. matching the # of FLOPs for inference over the dataset between the reduced fixed cutoff and dynamic cutoff models) is not as performant as using a dynamic cutoff model.
> > >
> > > >MD22 is a relatively simple dataset. I would suggest evaluating the method on more challenging datasets. For example, it would be helpful to test ORB and MACE on OMol.
> > >
> > > In Table 2, we show the training results on the challenging, foundational MatPES dataset [1]. Here is Table 2 copied for your convenience:
> > >
> > > | Model     | Metric | Fixed | Dynamic |
> > > |-----------|--------|-------|---------|
> > > | MACE      | Energy | 30    | 31      |
> > > |           | Forces | 173   | 190     |
> > > | NequIP    | Energy | 59    | 70      |
> > > |           | Forces | 167   | 168     |
> > > | Orbv3     | Energy | 67    | 68      |
> > > |           | Forces | 176   | 177     |
> > > | TensorNet | Energy | 39    | 43      |
> > > |           | Forces | 152   | 160     |
> > >
> > > During our response to Reviewer 93ox, we also trained a dynamic cutoff version of the 25.1 million parameter Orbv3 model on the OMol dataset:
> > >
> > > | Model                         | Energy Error (meV/atom) | Force Error (meV/Å) |
> > > |-------------------------------|--------------------------|----------------------|
> > > | Orbv3 OrbMol, fixed cutoff    | 2.7                      | 19.6                 |
> > > | Orbv3 OrbMol, dynamic cutoff  | 4.1                      | 21.8                 |
> > >
> > > >For some MLIPs that involve bond graphs, the construction of the bond graph and feature updates can be more memory-intensive. I suggest that the authors evaluate CHGNet and DPA3, and report both the accuracy and the memory reduction by using the proposed method in the paper.
> > >
> > > Due to the time constraints of the rebuttal acknowledge period, we will perform the additional experiments on CHGNet and DPA3 and include the results on the same training splits as the rest of the models in the camera-ready version of the manuscript. However, we would like to point out that we have already achieved strong results on the 4 SOTA models that we have implemented the dynamic cutoff on.
> > >
> > > Furthermore, in our Reply Rebuttal Comment to Reviewer 93ox, we show that **given fixed compute, a large model with a dynamic cutoff formulation achieves significantly lower error than a smaller model with a fixed cutoff formulation.** Please refer to that comment for more details. We believe there is a place in scientific discovery for both highly accurate, but slow MLIPs as well as less accurate, but extremely fast MLIPs. Dynamic cutoffs move the Pareto frontier forward in the class of less accurate, but extremely fast MLIPs.
> > >
> > > We would like to once again thank you for the rebuttal acknowledgement and hope we were able to completely address your concerns!
> > >
> > > [1] https://arxiv.org/abs/2503.04070

---

### Decision · Program_Chairs · 2026-04-30

**Decision:**

Accept (regular)

**Comment:**

The reviewers generally agree that the paper proposes a novel and technically meaningful idea: replacing fixed cutoffs in MLIPs with a smooth per-atom dynamic cutoff that preserves differentiability while reducing graph density. Reviewer 93ox was positive throughout and viewed the work as a sensible efficiency-oriented contribution, while reviewer eJnS was ultimately convinced by the rebuttal and raised the score from reject-leaning to accept after the authors clarified the role of the hard cutoff, robustness to hyperparameters and density variation, and the compute-vs-accuracy tradeoff. The rebuttals also addressed several concerns from reviewers F9HT and 1y1L, including comparisons against KNN constructions, clarification of periodic systems and force computation, additional training-time and memory tables, and stronger discussion of when pruned edges matter. Although some concerns remain — especially that the runtime-speedup analysis is still less convincing than the memory-reduction evidence, and that the method introduces a real accuracy/approximation tradeoff — I do not see a fatal flaw, and no reviewer identified prior work that substantially undermines the paper’s novelty. Overall, this seems like a useful and sufficiently original contribution whose practical value justifies acceptance, albeit near the margin.